# 4D structural biology–quantitative dynamics in the eukaryotic RNA exosome complex

Jobst Liebau [1] ✉, Daniela Lazzaretti [1] ✉, Torben Fürtges [2,3], Anna Bichler [1], Michael Pilsl [4], Till Rudack[2,3] ✉ & Remco Sprangers [1] ✉

Molecular machines play pivotal roles in all biological processes. Most structural methods, however, are unable to directly probe molecular motions. Here, we demonstrate that dedicated NMR experiments can provide quantitative insights into functionally important dynamic regions in very large asymmetric protein complexes. We establish this for the 410 kDa eukaryotic RNA exosome complex that contains ten distinct protein chains. Methyl-group and fluorine NMR experiments reveal site-specific interactions among subunits and with an RNA substrate. Furthermore, we extract quantitative insights into conformational changes within the complex in response to substrate and subunit binding for regions that are invisible in static cryo-EM and crystal structures. In particular, we identify a flexible plug region that can block an aberrant route for RNA towards the active site. Based on molecular dynamics simulations and NMR data, we provide a model that shows how the flexible plug is structured in the open and closed conformations. Our work thus demonstrates that a combination of state-of-the-art structural biology methods can provide quantitative insights into large molecular machines that go significantly beyond the well-resolved and static images of biomolecular complexes, thereby adding the time domain to structural biology.

Protein dynamics are tightly coupled with function[1–3]. Nuclear magnetic resonance (NMR) methods are particularly well suited to study dynamic processes in solution, at quasi atomic resolution and on a wide range of timescales[4–6]. Recent advances in sample preparation combined with NMR pulse-sequence and hardware design have made complexes over 100 kDa accessible to detailed solution NMR studies[7–11]. This thus opens up ample opportunities where NMR methods can complement static structural information obtained by e.g. single particle cryo-electron microscopy (cryo-EM) or in silico tools[12,13]. Specifically, NMR enables the study of transient interactions and dynamics on a wide range of timescales, which are crucial for the function of enzymes, yet difficult to analyze by static structural methods. NMR can thus add the time dimension to the three structural dimensions that static methods provide. Despite technological advances large and asymmetric complexes have eluded NMR investigation. Such complexes are substantially more challenging to study than large symmetric complexes, since symmetry gives rise to signal enhancement. At the same time, large, asymmetric protein assemblies are ubiquitous[14] and thus of interest to structural biology.

In the cytoplasm, the exosome (Fig. 1A) is involved in the canonical turnover of mRNA and in mRNA quality control; in the nucleus the complex degrades and processes a wide variety of RNA substrates[15,16]. The exosome is a modular molecular machine that consists of an inert, nonameric core (Exo9; 300 kDa). This core contains an essential central channel that is formed by six distinct RNase PH-like subunits (Rrp41, Rrp45, Rrp43, Rrp46, Mtr3, Rrp42) and a

[1]Department of Biophysics I, Regensburg Center for Biochemistry, University of Regensburg, Universitätsstraße 31, Regensburg, Germany. [2]Structural Bioinformatics Group, Regensburg Center for Biochemistry, University of Regensburg, Universitätsstraße 31, Regensburg, Germany. [3]Structural Bioinformatics Group, Regensburg Center for Ultrafast Nanoscopy, University of Regensburg, Universitätsstraße 31, Regensburg, Germany. [4]Structural Biochemistry Group, Regensburg Center for Biochemistry, University of Regensburg, Universitätsstraße 31, Regensburg, Germany. ✉e-mail: jobst.liebau@ur.de; daniela.lazzaretti@ur.de; till.rudack@ur.de; remco.sprangers@ur.de

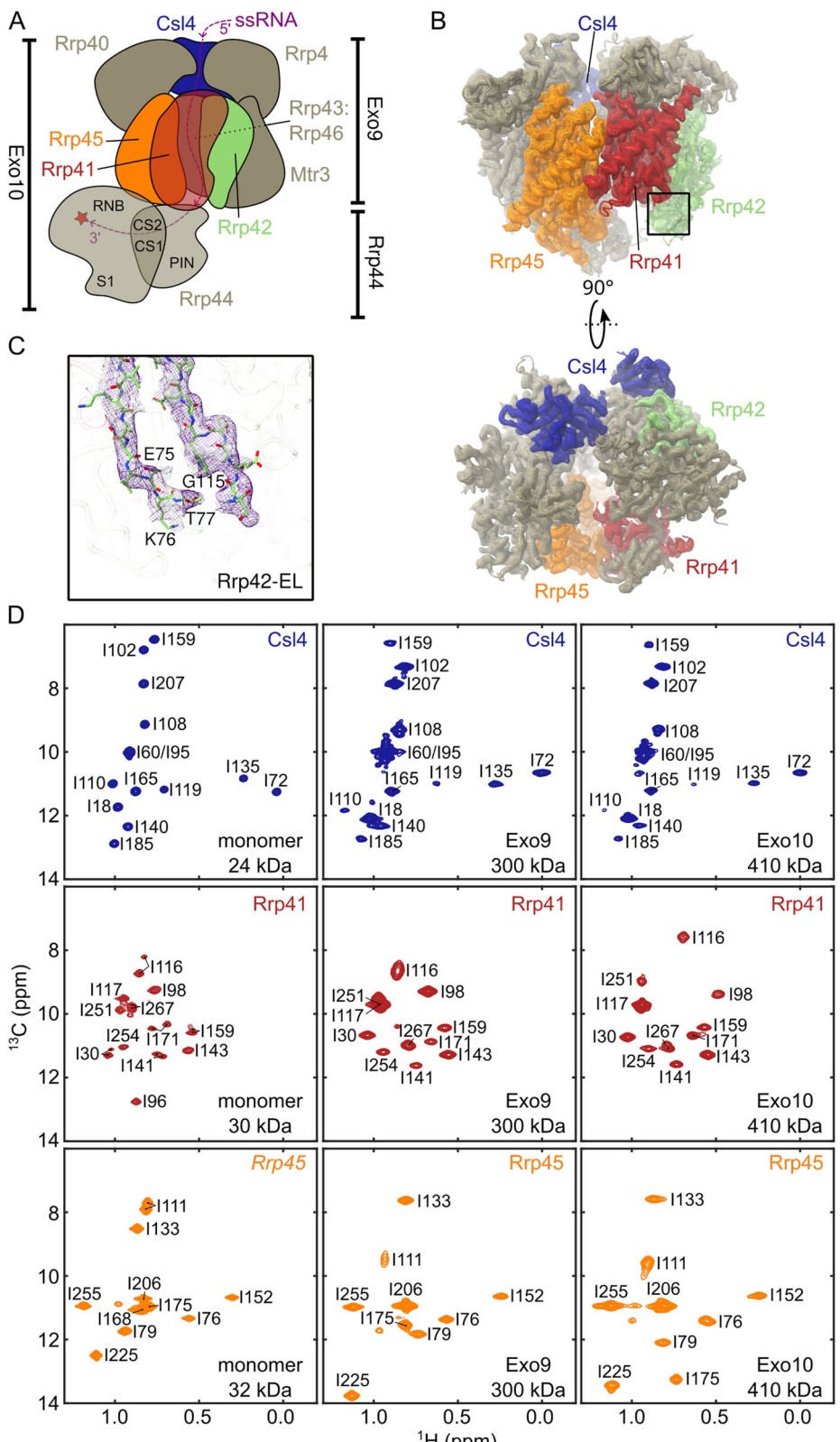

**Fig. 1 | Structure of ctExo9 and assignment of NMR spectra. A** Schematic depiction of Exo10. Throughout the text the Csl4, Rrp41, Rrp42 and Rrp45 subunits are colored blue, red, green and orange, respectively. The individual Rrp44 domains (PIN; pilT N-terminal, CS: cold-shock, RNB: RNA binding and S1) are labeled and the exonucleolytic site is highlighted with a star. The path of the RNA towards the active site is indicated with a purple line. **B** Side view (top) and top view (bottom) of the ctExo9 cryo-EM density map. **C** Zoom around the boxed region in panel B, top, that displays the cryo-EM density around the invisible section of the extended loop in Rrp42 (Rrp42-EL). **D** Ile-δ1 region of methyl-TROSY spectra for Csl4, Rrp41 and Rrp45 in the monomeric form (column 1) and when reconstituted into Exo9 (column 2) or Exo10 (column 3). Resonance assignments are indicated.

substrate entrance pore that is formed by three cap subunits (Csl4, Rrp4, Rrp40) that contain RNA binding domains[17]. Rrp41 and Rrp45 recruit the catalytic subunit Rrp44[18] to assemble the catalytically active decameric complex (Exo10; 410 kDa). Within Rrp44, the RNB domain harbors processive exonucleolytic activity, while the PIN domain can hydrolyze RNA in an endonucleolytic manner[19,20]. During catalysis, the 3′ end of a single-stranded RNA substrate is recruited by the cap subunits, threaded through the channel[21] and is finally presented to Rrp44 (Fig. 1A). Isolated exosomes degrade single-stranded RNA irrespective of the nucleotide composition[18]. Substrate specificity is conveyed by several compartment-specific co-factors associated with the complex[19,22]. Mutations in the exosome complex have been linked to multiple human diseases, underscoring its central functional importance[23]. In the past, static structures of the human[19,24,25] and yeast[26–33] exosome complexes have been reported that reveal its sub-unit organization and RNA interactions. The exosome of *Chaetomium thermophilum* exhibits improved thermal stability compared to exosome complexes from other organisms allowing for experiments to be conducted at up to 40 °C. In addition, *C. thermophilum* subunits can be expressed as monomers (or heterodimers) with sufficiently high yields and the complex can be reconstituted in vitro rendering it suitable for NMR study.

Here, we demonstrate that by combining recent developments in labeling strategies and experimental design, very large, fully asymmetric protein complexes such as the RNA exosome are amenable to NMR study. We show that a wide range of NMR experiments, informing on interactions, structure and dynamics can be applied to such systems. In combination with single particle cryo-EM, X-ray crystallography and molecular dynamics (MD) simulations unique insights into dynamic properties of protein complexes can then be obtained. This approach thus facilitates studies of large and fully asymmetric protein complexes that go substantially beyond mere static snapshots.

## Results and discussion

### Static structures of the ctExo9 complex
We determined the structure of Exo9 from the eukaryotic thermophile *C. thermophilum* (ctExo9) by X-ray crystallography to 3.8 Å resolution (Supplementary Fig. 1B, C, Supplementary Table 1A) and by cryo-EM to 3.2 Å resolution (Fig. 1B, Supplementary Fig. 1A, C, Supplementary Fig. 2, Supplementary Table 1B). The general architecture of ctExo9 is identical to that of yeast and human exosome complexes[19,24–32]. Despite this wealth of structural information, several disordered regions are invisible in these structures. In particular, an extended loop region in Rrp42 (Rrp42-EL) and the entry loop in Rrp41 are largely unresolved, while the shorter exit loop of Rrp41 is only partially visible (Fig. 1C, and Supplementary Figs. 1A, D, E). A priori, these invisible regions cannot be considered functionally unimportant, as disordered regions are often directly involved in biological function[3,34,35].

### NMR assignments in the exosome complex
To obtain insights complementary to the static structures, we turned to methyl-based NMR spectroscopic methods[10]. Such approaches have been successfully applied to large, highly symmetric protein assemblies with molecular weights of up to 1 MDa[36–39] and to single-chain proteins of up to 100 kDa[40]. In that light, the eukaryotic exosome complex is significantly more challenging to study, as it contains ten distinct protein chains with a total molecular weight of almost half a megadalton. To render the exosome complex visible to NMR spectroscopy, we employed a labeling scheme, in which one subunit at a time was labeled with NMR-active Ile-δ1[$^{13}$CH$_3$] and Met-ε1[$^{13}$CH$_3$] methyl groups in an otherwise fully deuterated background (IM-labeling) (Supplementary Fig. 3). Methyl resonance assignments were obtained by exploiting a divide-and-conquer strategy, where we first assigned resonances in the monomeric subunits Csl4, Rrp41 and Rrp45 (Fig. 1D, and Supplementary Fig. 4). These assignments were then

transferred to the Exo9 and Exo10 complexes, assisted by numerous point mutants (Supplementary Table 6C, exemplified in Supplementary Fig. 5). The assignment completion of the Ile-δ1 resonances was close to 90% (Supplementary Table 2) providing a set of NMR probes that can report on interactions and dynamics and that are well distributed over the complex.

### Interactions between Exo9 and Rrp44
Based on chemical shift perturbations (CSPs), site-specific insights into intermolecular interactions can be obtained. Chemical shifts of a number of resonances in the ring subunits Rrp41 and Rrp45 differ significantly between the Exo9 and Exo10 complexes, whereas resonances in Csl4 were unaffected by the addition of Rrp44 (Fig. 1D, and Supplementary Fig. 6). These observations are in agreement with existing structural information for yeast and human exosomes that show that Rrp44 is recruited to the Exo9 complex by Rrp41 and Rrp45[19,28–31]. Our data thus reveal that interactions between ctExo9 and ctRrp44 are conserved and that methyl-TROSY methods can be exploited to identify interaction interfaces in large asymmetric eukaryotic assemblies.

The methyl-TROSY methods that we deployed are blind in regions that are devoid of Ile or Met residues. To also investigate such regions, we turned to $^{19}$F NMR methods that were recently shown to be excellent tools to study interactions and dynamics on a broad range of timescales[41–44], even for larger complexes[45–48]. First, we employed amber codon suppression to introduce a 4-trifluoromethyl-L-phenylalanine (tfmF) into Rrp41 at position D113 (Rrp41$^{D113tfmF}$). Based on our structures, this position is located next to a partially structured loop (exit loop) that faces the Rrp44 interaction interface of Exo9 and that lines the exit site of the RNA channel (Supplementary Figs. 1A, D). Upon addition of Rrp44 the resonance of Rrp41$^{D113tfmF}$ shifts, demonstrating its spatial proximity to Rrp44 (Supplementary Fig. 7A). Second, we introduced a tfmF label at position Q86 in Rrp41 (Rrp41$^{Q86tfmF}$) that is located in an extended loop close to the cap subunits and not visible in any of the structures (entry loop, Supplementary Figs. 1A, D). This resonance is not affected by the addition of Rrp44, in agreement with a remote location of the entry loop from Rrp44 (Supplementary Fig. 7B). To probe if the invisible entry loop approaches the entry site of the RNA channel, we assembled an exosome complex, in which the cap subunit Csl4 was labeled with a paramagnetic 2,2,6,6-tetra-methylpiperidine-1-oxyl (TEMPO) spin-label at position E130 (Csl4$^{C122S, E130C-TEMPO}$, Supplementary Fig. 1C). Rrp41$^{Q86tfmF}$ proved to be too remote to be affected by the Csl4 spin-label. However, Rrp41$^{G71tfmF}$, for which the flourine label is located in the center of the entry loop, displayed fluorine paramagnetic relaxation enhancements (PREs, $\Gamma$) that are a direct reporter of the distance between the Csl4 spin-label and Rrp41$^{G71tfmF}$. The spin-label gives rise to enhanced $R_1$ relaxation rates (Supplementary Fig. 8, Supplementary Table 3A). $R_2$ relaxation rates are also enhanced as judged from a qualitative comparison of diamagnetic versus paramagnetic spectra (Supplementary Fig. 8). This establishes that the Rrp41 entry loop is located at the entry site of the RNA channel. To probe if the entry (Rrp41$^{Q86tfmF}$) and exit (Rrp41$^{D113tfmF}$) loops undergo motions on the micro- to millisecond timescale we measured fluorine CPMG relaxation dispersion experiments (Supplementary Fig. 9) that reveal no signs of chemical exchange, indicating that both loops move on a fast (≪ ms) timescale in solution.

### RNA threads through the exosome channel
To investigate interactions between the exosome and RNA in solution, we first assessed methyl CSPs in the subunit-specific IM-labeled Exo9 and Exo10 complexes. To prevent RNA degradation, we reconstituted Exo10 with inactive Rrp44$^{D168N-D536N}$ in all NMR experiments conducted in the presence of RNA. These data reveal that Csl4, Rrp41 and Rrp45 all interact with the substrate (Fig. 2A). In Csl4, CSPs are most pronounced in the S1 domain (residues 98-178) indicating

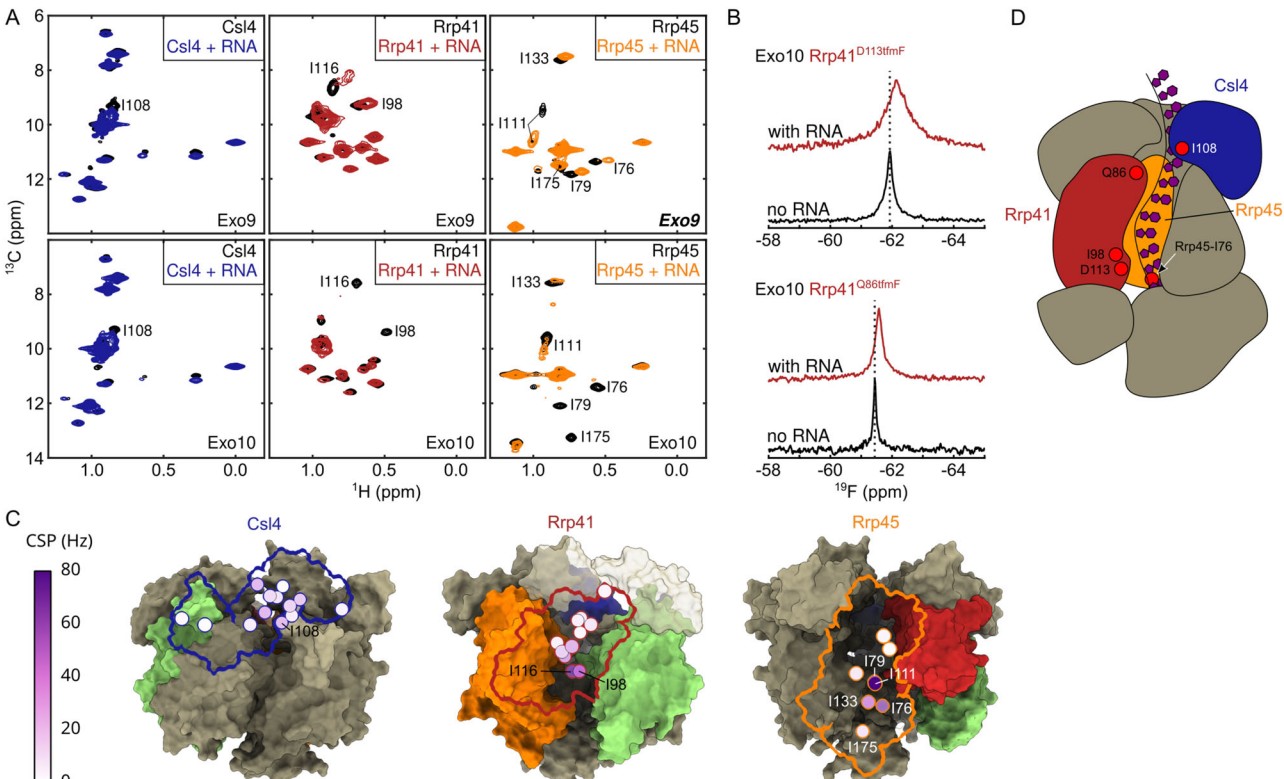

**Fig. 2 | RNA interaction in the exosome. A** Ile-δ1 region of methyl-TROSY spectra in the absence (black) and presence of RNA for Csl4 (blue), Rrp41 (red) and Rrp45 (orange) reconstituted into Exo9 (top) and Exo10 (bottom). Residues that show strong CSPs are labeled in the spectra. **B** 1D [19]F spectrum of Rrp41[D113tfmF] (top; exit loop) and Rrp41[Q86tfmF] (bottom; entry loop) reconstituted into Exo10 with (red) and without (black) RNA. The dashed line indicates the center of the resonance for Exo10 without RNA. Both loops are affected by RNA interactions. **C** RNA-induced

CSPs of Ile-δ1 for Exo9 plotted onto the cryo-EM structure with the same coloring scheme as in panel A. Rrp42 is in green. For clarity only the outline of the subunit that is NMR active is shown. **D** Schematic depiction of Exo10 showing a tentative RNA path through the exosome channel. The coloring scheme is as in panel A. Rrp42 is omitted for clarity. Positions of a number of RNA-interacting residues are indicated.

direct interactions with the linear RNA substrate (Fig. 2C, D). This is expected since S1 domains have been implicated in RNA binding[49,50]. In Rrp41 and Rrp45, resonances of residues that line the channel are affected by RNA (Fig. 2A, C), consistent with RNA being threaded through the channel (Fig. 2D). Additionally, [19]F NMR data confirm the involvement of both the Rrp41 entry and exit loops in RNA interactions (Fig. 2B, Supplementary Fig. 8C). Moreover, for the entry loop, we observe a reduction of the PRE effect that is caused by Csl4[C122S, E130C-TEMPO] upon addition of RNA (Supplementary Fig. 8A, B). This indicates that the dynamic entry loop is displaced away from Csl4 when RNA enters the exosome barrel (Supplementary Fig. 8D). Based on previous RNA interaction studies and alignments of the ctRrp41 and ctRrp45 sequences with corresponding archaeal, yeast and human sequences (Supplementary Fig. 10), we establish that the RNA coordination via positively charged residues inside the channel is conserved among those species[18,51].

### RNA displaces a channel exit loop

Next, we investigated the dynamics and function of an extended loop region in the Rrp42 subunit, Rrp42-EL (Supplementary Fig. 11), that is unresolved and thus invisible in both the X-ray and cryo-EM structure (Fig. 1C). To obtain insights into the location of Rrp42-EL in the exosome complex, we engineered a double mutant, Rrp42[C59S, A106C], in which the single wild-type Cys residue is replaced by a Ser residue (C59S) and a new Cys residue is incorporated into Rrp42-EL (A106C). Subsequently, we attached a TEMPO spin-label to Rrp42-EL (Rrp42[C59S, A106C-TEMPO]) and reconstituted this subunit together with IM-labeled Csl4, Rrp41 or Rrp45 into Exo9 and Exo10 complexes. Csl4

resonances are not affected by the spin-label (Fig. 3A, E), indicating that Rrp42-EL does not approach the cap subunit Csl4. In contrast, a number of Rrp41 and Rrp45 resonances display substantial PRE effects ($I_{para}/I_{dia} < 1$) in Exo9 and Exo10 (Fig. 3B, C, F, G, and Supplementary Fig. 12). The affected residues face the exit site of the Exo9 channel, and PREs are stronger in the Exo10 complex than in the Exo9 complex indicating that the conformation of invisible Rrp42-EL is affected by Rrp44. In the presence of RNA substrate, PRE effects in Rrp45 are obliterated (Fig. 3D, and Supplementary Fig. 12), indicating that RNA displaces the loop away from Rrp45. Based on that, we conclude that Rrp42-EL adopts two conformations: one, in which it is proximal to the channel and which is stabilized by Rrp44 (closed) and another, in which it is distant from the channel and which is adopted in the presence of RNA (open).

### Rrp44 and RNA modulate Rrp42-EL dynamics

To obtain direct insights into the dynamics of the invisible Rrp42-EL, we labeled A106C with 3-Bromo-1,1,1-trifluoro-acetone (BTFA) to form Rrp42[C59S, A106C-TFA] (Supplementary Fig. 13), which retains ribonucleolytic activity of the Exo10 complex (Supplementary Fig. 14). Within the Exo9 complex, Rrp42[C59S, A106C-TFA] displays one fluorine resonance (Fig. 4A); however, CPMG relaxation dispersion measurements (Fig. 4B, and Supplementary Fig. 15B) reveal the presence of a second, minor conformation and, consistently, chemical exchange saturation transfer (CEST) experiments show an asymmetric CEST dip (Supplementary Fig. 15A). Addition of single-stranded RNA results in a minor shift of the fluorine resonance frequency (Fig. 4A) and restricts the motions of the loop considerably as evidenced by an attenuated

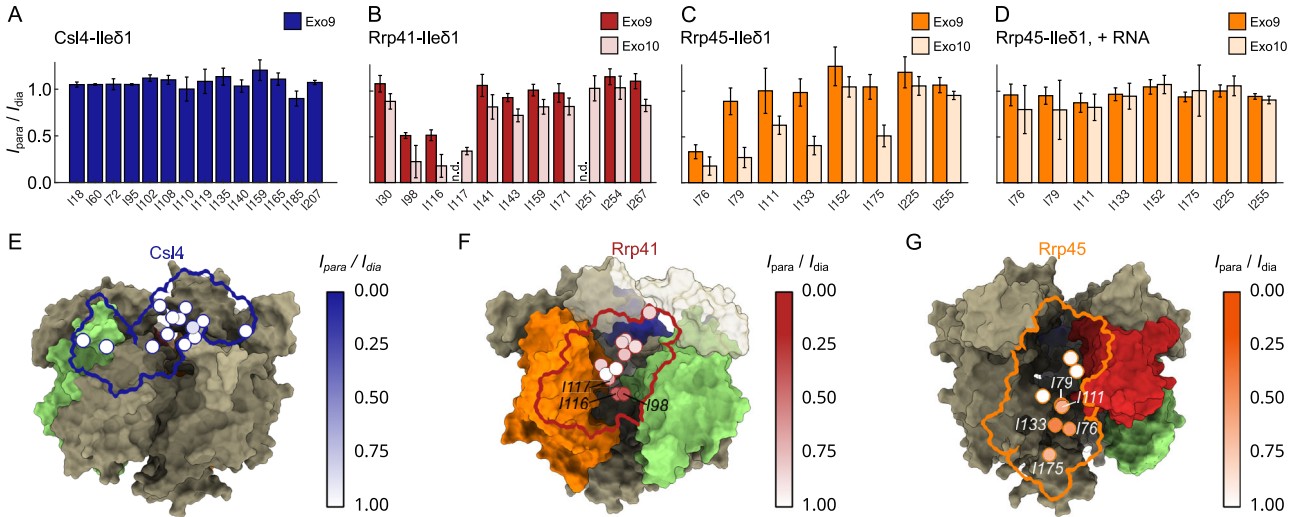

**Fig. 3 | Localization of Rrp42-EL.** PRE effects of Rrp42$^{C59S, A106C-TEMPO}$ on Ile-δ1 of (**A**) Csl4 in Exo9, (**B**) Rrp41 in Exo9 (red) and Exo10 (light red), (**C**) Rrp45 in Exo9 (orange) and Exo10 (light orange) without RNA and (**D**) Rrp45 in Exo9 (orange) and Exo10 (light orange) with RNA. n.d.: value not determined due to signal overlap. Data were obtained from one measurement. Error bars are derived from the signal-to-noise ratio in the NMR spectrum and represent ±1 SD. Ile-δ1 PREs of (**E**) Csl4 (in Exo9), (**F**) Rrp41 (in Exo10) and (**G**) Rrp45 (in Exo10) plotted onto the cryo-EM structure. Csl4 is shown in blue, Rrp41 in red, Rrp45 in orange and Rrp42 in green. Note, that Rrp42-EL is not visible in the structure (see Fig. 1C). For clarity only the outline of the subunit that is NMR active is shown. Residues that show strong PREs are labeled.

relaxation dispersion profile and a symmetric CEST dip (Fig. 4B, and Supplementary Fig. 15C, D).

To obtain additional information on the localization and dynamics of this invisible loop, we attached a spin-label to position A106C (Rrp42$^{C59S, A106C-TEMPO}$) and determined its PRE effect on the $^{19}$F resonance in Rrp41$^{D113tfmF}$ (Fig. 4C, Supplementary Table 3B). In the absence of RNA, sizable PRE effects are visible for $R_1$ and $R_2$ relaxation rates, indicating that Rrp42-EL and the Rrp41 exit loop come within less than ~10 Å from each other. This is in agreement with the complementary methyl-TROSY data (Fig. 3B), where spin-labeled Rrp42-EL caused PRE effects close to the exit loop of Rrp41 (e. g. I98 and I116). Upon addition of substrate, the fluorine PRE effects are abolished (Fig. 4C, H), which implies that the Rrp41 exit loop and the invisible Rrp42-EL move apart.

We next turned to the Exo10 complex, in which the fluorine label in Rrp42-EL displays a second downfield-shifted resonance implying the formation of a second, long-lived conformation (Fig. 4D). This second conformation is induced by the C-terminal RNB-S1 domains of Rrp44, as Exo10 complexes that only contain the Rrp44 PIN domain or the PIN domain plus the two cold-shock (CS) domains fail to stabilize the second conformation (Supplementary Fig. 16). Structurally, a direct interaction between Rrp42-EL and the Rrp44-RNB-S1 domains is unlikely based on known structures of the human and *S. cerevisiae* Exo10 complexes (Supplementary Fig. 17). It is, however, plausible that the CS domains are brought into close spatial proximity of Rrp42-EL when Rrp44-RNB-S1 interacts with Exo9 as also indicated by a model of ctExo10 (see below).

The dynamics of Rrp42-EL in the Exo10 complex can be directly probed using CEST (Fig. 4E, and Supplementary Fig. 15E) and longitudinal exchange (EXSY) (Supplementary Fig. 15F) experiments. Since EXSY experiments exclusively probe motions on the slow NMR timescale, we conclude that the dynamics of Rrp42-EL are significantly slowed down by Rrp44. $^{19}$F PRE effects show that Rrp42-EL in the Exo10 complex is still in close proximity to the Rrp41 exit loop (Fig. 4F), which corroborates the methyl-TROSY data (Fig. 3B). Upon addition of RNA to Exo10, the downshifted $^{19}$F resonance and PRE effects disappear and there are no indications for motions in the CEST profiles (Fig. 4D–F, J, and Supplementary Fig. 15G, H) implying that Rrp42-EL is fully in the open state. Rrp42-EL likely adopts multiple inter-converting conformations in the open state as evidenced by a weak relaxation

dispersion profile of Exo10 Rrp42$^{C59S, A106C-TFA}$ in the presence of RNA (Supplementary Fig. 15H) and a shoulder of the open-state resonance in the absence of RNA (Fig. 4D, and Supplementary Fig. 15F, inset).

### Quantification of Rrp42-EL dynamics

Interestingly, we observe that upon addition of Rrp44 to the apo Exo9 complex $\Gamma_2$ rates are enhanced (spectra in Fig. 4C, F; indicating that Rrp42-EL moved towards Rrp41 upon formation of the Exo10 complex), while $\Gamma_1$ rates are diminished (inversion recovery plots in Fig. 4C, F, supplementary table 3B; indicating that Rrp42-EL moved away from Rrp41 upon formation of the Exo10 complex). This apparent contradiction can be explained by the differential dependence of $\Gamma_1$ and $\Gamma_2$ on fast timescale motions. As noted before[52,53] decreased order parameters ($S^2$) and increased internal motions ($\tau_i$) can result in enhanced $\Gamma_1$ rates, whereas $\Gamma_2$ rates are largely unaffected by motions that are faster than the rotational correlation time (Eq. 9, Supplementary Fig. 19). Our data thus imply that the invisible Rrp42-EL is more rigid in the closed conformation than in the ensemble of open conformations. At the same time the closed conformation is more prominently populated in the Exo10 complex than in the Exo9 complex.

To quantitatively assess the dynamics of the open-closed equilibrium and to obtain insights into order parameters of Rrp42-EL in the two states, we globally fitted a two-site exchange model to the dynamics experiments (CEST, RD, EXSY) of apo Exo9 and Exo10, and to $^{19}$F PRE experiments in the Exo9 and Exo10 complexes in the absence and presence of RNA (Supplementary Fig. 15, Supplementary Fig. 18, Supplementary Fig. 20, Supplementary Table 4). We assumed that the chemical shifts, local correlation times and order parameters of the open and closed states are the same in the Exo9 and Exo10 complexes. We further assumed that Rrp41$^{D113tfmF}$ adopts only one conformation. For diamagnetic Exo10 Rrp42$^{C59S-A108C-TEMPO}$ Rrp41$^{D113tfmF}$ this is clearly not correct since the resonance displays a shoulder, implying that parameters extracted for this sample are population-weighted averages of the two conformations of Rrp41$^{D113tfmF}$ in Exo10. We deconvoluted 1D spectra and $T_1$ data for this sample. While the shoulder peak (75% population) shows faster $R_2$ relaxation, both components have identical $R_1$ relaxation rates (Supplementary Figs. 18 E–G).

The global analysis of dynamics and PRE data revealed that in Exo9 the invisible Rrp42-EL adopts the closed (open) conformation to

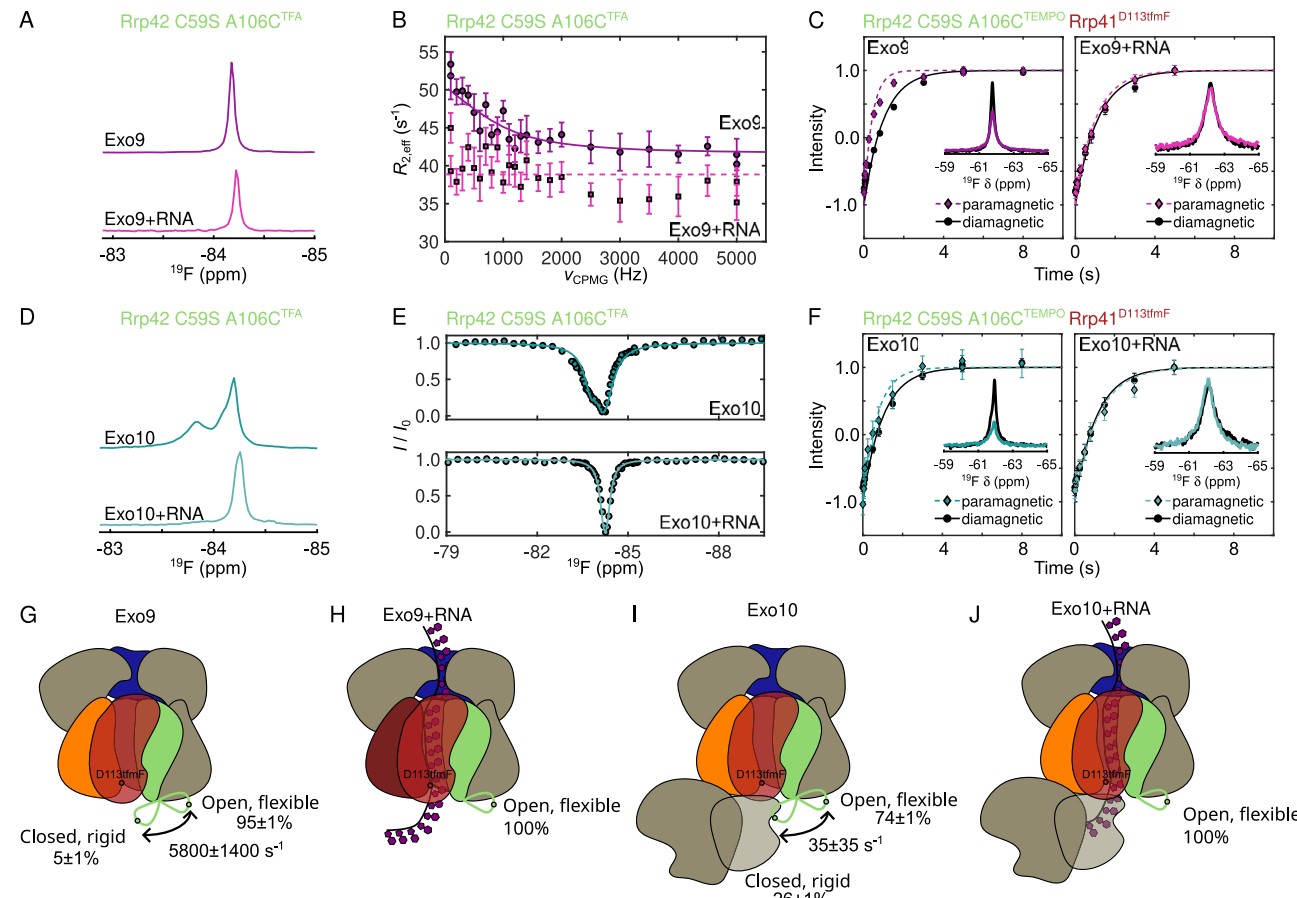

**Fig. 4 | Dynamics of Rrp42-EL. A** $^{19}$F spectra of Rrp42$^{C59S, A106C-TFA}$ in Exo9 without (top, purple) and with (bottom, pink) RNA and (**B**) corresponding RD profiles. **C** Inversion recovery experiments (curves) and $^{19}$F 1D spectra (insets) for Rrp41$^{D113tfmF}$ in Exo9 Rrp42$^{C59S, A106C-TEMPO}$ without (left) and with (right) RNA for paramagnetic (in color) and diamagnetic (in black) samples. (**D**) $^{19}$F spectra of Rrp42$^{C59S, A106C-TFA}$ in Exo10 without (top, teal) and with (bottom, light teal) RNA and (**E**) corresponding CEST profiles, acquired for $B_1 = 25$ Hz. **F** Inversion recovery experiments (curves) and $^{19}$F 1D spectra (insets) for Rrp41$^{D113tfmF}$ in Exo10 Rrp42$^{C59S, A106C-TEMPO}$ without (left) and with (right) RNA for paramagnetic (in color)

and diamagnetic (in black) samples. **G, J** Schematic depiction of Rrp42-EL dynamics in Exo9 (**G**) without and (**H**) with RNA and in Exo10 (**I**) without and (**J**) with RNA. Rrp41 is in red, Rrp45 in orange, Csl4 in blue and Rrp42 in green. The green loop depicts Rrp42-EL, the green dot A106C$^{TFA/TEMPO}$ and the red dot Rrp41$^{D113tfmF}$. Fits for panels C and F were obtained using Eq. 1. Fits for panels B and E were obtained using the model described in the methods. Error bars in B, C and F are derived from the signal-to-noise ratio in the NMR spectrum and represent ±1 SD. Errors in **G** and **I** are ±1 SD and obtained as described in the "Method section".

$5 \pm 1\%$ ($95 \pm 1\%$) and that the open to closed transition takes place at a rate ($k_{ex} = k_{open \to closed} + k_{closed \to open}$) of $5800 \pm 1400$ s$^{-1}$ (Fig. 4G). In the Exo10 complex the population of the closed conformation is significantly higher ($26 \pm 1\%$), whereas the exchange rate is reduced to $35 \pm 35$ s$^{-1}$ (Fig. 4I). The order parameter ($S^2$) of the open conformation (~ 0.1) is significantly lower than of the closed conformation (~ 0.7), revealing that the loop in the open state is highly flexible, whereas it is stably fixed to the rest of the exosome complex in the closed state. In the presence of RNA, the open conformation in the Exo9 and Exo10 complex is occupied to 100% (Fig. 4C, F, H, J), indicating that Rrp42-EL is fully displaced by substrate RNA.

## Structural insights into Rrp42-EL

The Rrp42 extended loop is invisible in the static cryo-EM and crystal structures (Fig. 1). To obtain further structural and dynamic insights into Rrp42-EL in the closed and open conformation, we exploited molecular dynamics (MD) simulations of the Exo9 complex in aqueous solution. To initiate the MD simulations of the open state, all missing loops of the here obtained cryo-EM structure were modeled using the ColabFold implementation of AlphaFold2[13]. The closed state was obtained by interactively modeling Rrp42-EL into the unoccupied cavity near Rrp41, a location that agrees with our NMR data (Figs. 3, 4). Our MD simulations reflect that the open and the closed conformations are

energetically stable (Fig. 5A, and Supplementary Fig. 21). The nanosecond timescale mobility of Rrp42-EL in the closed state is clearly reduced compared to the open state, as monitored by the C$^\alpha$ root mean square deviation (RMSD) of Rrp42-EL (Fig. 5B). This reduced flexibility within the MD simulations is fully consistent with the NMR order parameter ($S^2$) analysis (Fig. 4, and Supplementary Fig. 18). Moreover, in the open state Rrp42-EL is remote to Rrp41 forming very few intersubunit contacts restricted solely to Mtr3 (Fig. 5C). In contrast, in the closed state Rrp42-EL forms numerous contacts with Mtr3, Rrp43 and Rrp45 (Fig. 5C), in agreement with the PRE experiments (Fig. 3). In the closed conformation, the Rrp42-A106 C$^\alpha$ atom remains in close distance to the Rrp41-D113 C$^\alpha$ atom, whereas this distance is much longer in the open state (Supplementary Fig. 22, Supplementary Movie 1), which agrees qualitatively with the $^{19}$F PRE data (Fig. 4, and Supplementary Fig. 18). Interestingly, AlphaFold-predicted α-helical elements (residues K82-A94 and A106-N112) of Rrp42-EL begin to partially unfold during the simulation, indicating flexibility within the secondary structure (Supplementary Fig. 23). A comparison between the representative MD simulation structures complemented with Rrp44 from *S. cerevisiae* and the human Exo10 complex suggest that Rrp44 impacts the transition of Rrp42-EL between the open and closed conformation (Supplementary Fig. 24). The observed impact of Rrp44 agrees with what is expected from exosome structures of other organisms (Supplementary Fig. 17)

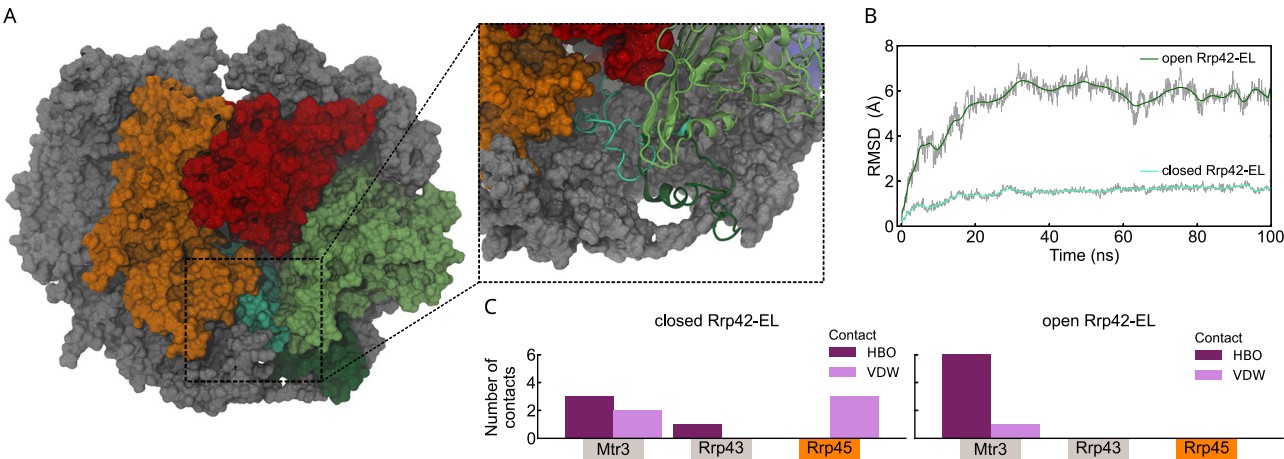

**Fig. 5 | MD simulations of Rrp42-EL in the closed and open state. A** Overlay of the representative structures from MD simulations of the completed Exo9 complex comparing the closed and open state of Rrp42-EL. The enlargement illustrates the distinct open (dark green) and closed (cyan) conformation of Rrp42-EL. Rrp41 is in red, Rrp45 in orange and Rrp42 in green. **B** Local structural dynamics of Rrp42-EL in the closed (cyan) and open (dark green) state revealed by the RMSD of the $C^{\alpha}$ carbons of only Rrp42-EL within the MD simulation. The RMSD is smoothed by a Bézier curve. **C** Interaction network analysis between the Exo9 subunits and Rrp42-EL in the closed (left) and open (right) conformation. Shown are the number of hydrogen bonds (HBO, violet) and van-der-Waals contacts (VDW, pink) between Rrp42-EL and the indicated subunits.

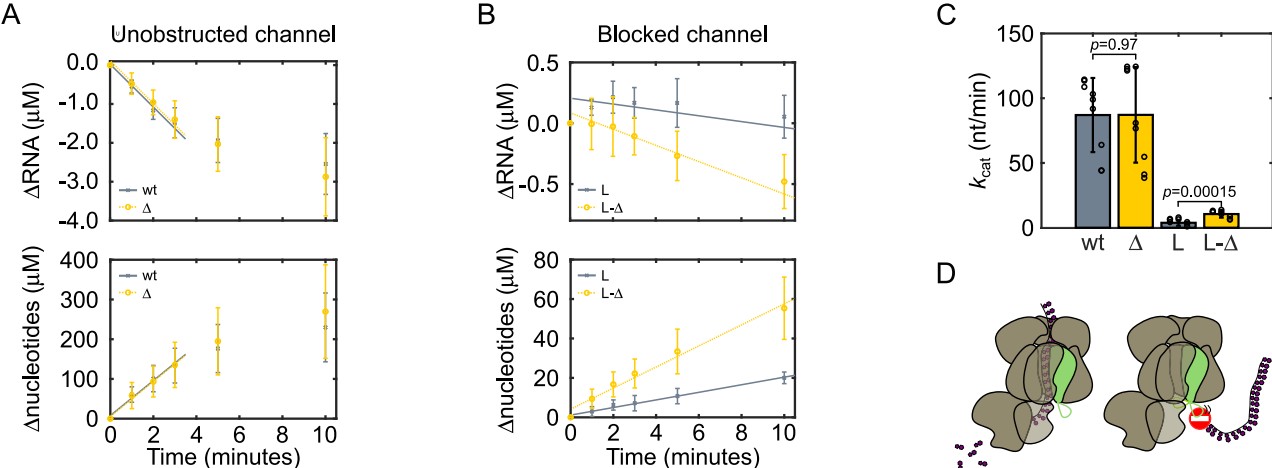

**Fig. 6 | Rrp42-EL blocks an aberrant RNA access path.** RNA activity assays for (**A**) wtExo10 (wt, gray crosses, solid line) and Exo10 Rrp42$^{\Delta93\text{-}125}$ (Δ, yellow circles, dashed line) and (**B**) for channel-blocked Exo10 Rrp45-L (L, gray crosses, solid line) and Exo10 Rrp45-L Rrp42$^{\Delta93\text{-}125}$ (L-Δ, yellow circles, dashed line). The lines are global linear fits to the linear activity regime. For an extended analysis of alternative linear regimes see Supplementary Fig. 26. **C** Catalytic activity of exosome constructs. *P* values are derived from a two-sided, paired-sample *t*-test. wt = wtExo10, Δ = Exo10 Rrp42$^{\Delta93\text{-}125}$, L = Exo10 Rrp45-L, L-Δ = Exo10 Rrp45-L Rrp42$^{\Delta93\text{-}125}$. Data were obtained from three biological replicates with three technical repeats. Error bars represent ±1 SD. **D** Rrp42-EL allows on-path RNA to access Rrp44 for degradation (left) but blocks a direct access path towards Rrp44 (right).

and with our NMR experiments (Fig. 4, and Supplementary Fig. 16). Finally, an analysis of the RNA channel within the representative MD simulation structures illustrates that the closed conformation of Rrp42-EL blocks the exit site of the RNA channel in the exosome core, whereas the RNA exit channel is unobstructed in the open state (Supplementary Fig. 25). Also in that regard, the MD and NMR data are fully consistent and explain that the closed state of Rrp42-EL is not observed in the presence of an RNA substrate (Fig. 4, and Supplementary Fig. 18).

### Rrp42-EL blocks an aberrant RNA path

The observation that Rrp42-EL can dynamically interact with the channel exit of the exosome raises the question of whether this is functionally relevant. To address this, we studied the activity of the Exo10 complex in the presence of full-length Rrp42 and with a version, in which Rrp42-EL was deleted (Rrp42$^{\Delta93\text{-}125}$). The RNA degradation rate in Exo10 is unaffected by the truncation of Rrp42 (Fig. 6A, C, and

Supplementary Fig. 26, Supplementary Table 10B), which means that the loop displacement by substrate RNA comes at a low energetic cost. In the canonical substrate route, RNA threads through the Exo9 channel; it has, however, been shown that RNA can employ alternative paths to the Rrp44 active site that bypass the Exo9 channel[33,54]. To assess if RNA can access Rrp44 via a direct path, we introduced an extension into a channel-lining loop of Rrp45, termed Rrp45-L, that has previously been shown to block the exosome channel in *S. cerevisiae*[17,54] (Supplementary Fig. 27). We observed that Rrp45-L reduces the activity of the exosome considerably, to ~4% of wild-type activity (Fig. 6B, C, and Supplementary Fig. 26, Supplementary Table 10B), confirming that the through-channel path is the major route that the RNA substrate employs. Next, we combined channel-blocked Exo10 Rrp45-L with the Rrp42$^{\Delta93\text{-}125}$ mutant, in which Rrp42-EL is deleted. Interestingly, we find that the activity in this exosome complex is partially recovered (Fig. 6B, C, Supplementary Fig. 26,

Supplementary Table 10B). Based on that we conclude that Rrp42-EL functions as a barrier that blocks an aberrant direct access path to the Rrp44 active site (Fig. 6D). Rrp42 in *C. thermophilum* thus contains a previously unidentified flexible one-way-plug that readily allows for passage of substrate RNA via the through-channel route but that prevents access to the Rrp44 active site via an aberrant direct path. The latter would result in the potentially detrimental ability of the exosome to degrade substrates that are not selected for processing or degradation by accessory factors, which interact with the cap subunits of the exosome complex.

We hypothesize that the observed population equilibrium and dynamics of Rrp42-EL (Fig. 4, and Supplementary Fig. 15) are fine-tuned to achieve a compromise between two conflicting objectives: Rrp42-EL in the closed state obstructs the path of channel-bound RNA to the active site. In order for this energetic barrier to not be rate-limiting, it should be as low as possible, i. e. the open state should be populated as much as possible. On the other hand, if the open state was to be fully populated, this would allow RNA to access the catalytic site through an aberrant path. To avoid this, the closed state should be populated as much as possible. An optimal balance thus entails a population equilibrium and dynamics that obstruct the aberrant path, while keeping the energy barrier for channel-bound RNA to pass the constriction site as low as possible.

As shown in Supplementary Fig. 11, Rrp42-EL is not conserved across species, suggesting that its specific role might be limited to the exosome of *C. thermophilum*. However, long disordered segments occur frequently in eukaryotes; the entrance loop of Rrp41 constitutes one additional example that we discussed in Supplementary Fig. 8. By characterizing Rrp42-EL we here describe one mechanism—a flexible plug that prevents aberrant access—that long disordered segments can fulfill. This may be a recurring theme in other organisms and thus of broader biological significance.

The resolution revolution in cryo-electron microscopy[55] and the remarkable performance of structure prediction algorithms have substantially increased insights into the relationship between protein structure and function. However, those methods provide static structural snapshots and often lack information on loop regions, protein dynamics and transient interactions, all of which may be crucial for protein function. Here, we demonstrate that dedicated NMR methods, such as CEST, EXSY, relaxation dispersion and PRE experiments and a combination of methyl-TROSY and $^{19}$F NMR, together with molecular dynamic simulations can complement static structural information, even in large, fully asymmetric eukaryotic molecular machines. In particular, we showed that quantitative and functionally important information can be obtained for regions that are invisible in structures derived from cryo-EM and X-ray crystallography. We demonstrate this for an entrance loop of the exosome and for Rrp42-EL, both of which are not visible in static structural snapshots. The dynamics of Rrp42-EL are altered by Rrp44, which slows down exchange dynamics by two orders of magnitude and increases the population of the closed state, and by RNA, in the presence of which only the open state is populated. In addition a combination of MD, methyl and $^{19}$F NMR allowed us to characterize structural features of Rrp42-EL. The observed dynamics are of functional importance, since Rrp42-EL seals an aberrant access path to the catalytic site, while being sufficiently dynamic to allow passage of properly inserted RNA, a conclusion that could not have been reached from static structures alone. Since large asymmetric or transiently formed complexes play a key role in virtually all aspects of molecular biology, we envision that the strategies to study large complexes by NMR and MD we laid out here will be of future importance to gain a deeper understanding of how protein structure, interactions and dynamics relate to function. We are convinced that the approach described here will spark further studies that facilitate the transition from 3D to 4D structural biology.

## Methods

### Molecular cloning

Codon-optimized constructs of the ten subunits of the *Chaetomium thermophilum* exosome (see Supplementary Table 5) were obtained from GenScript, cloned into pETM-11 vectors and expressed with an N-terminal hexahistidine-tag, a tobacco edge virus (TEV) cleavage site and a kanamycin resistance cassette. For monomeric Rrp43 and Rrp42$^{\Delta93\text{-}125}$, expression yields were low but yields could be improved by co-expression with Rrp46 or Mtr3, respectively, from bicistronic constructs, in which the downstream gene (*rrp46* or *mtr3*) did not code for a hexahistidine tag or a TEV cleavage site. Tight binding of Rrp43 to Rrp46 and of Rrp42$^{\Delta93\text{-}125}$ to Mtr3 allowed for co-purification of the subunits. In order to reduce the number of purifications required, this approach could also be used for wtMtr3-Rrp42 and wtRrp41-Rrp45, where appropriate, even though the monomeric proteins provided sufficient yields.

For cryo-EM and X-ray crystallography, Exo9 was expressed from a polycistronic pETM-11 plasmid that contained genes coding for all 9 subunits in the order: *rrp40-csl4-rrp4-rrp46-rrp43-mtr3-rrp42-rrp41-rrp45*. Only Rrp40 was expressed with an N-terminal hexahistidine-tag and a TEV cleavage site.

Point mutations, inserts and deletions of the original constructs were obtained using site-directed mutagenesis. Primers are listed in Supplementary Tables 6A (assignment mutants) and 6B (other constructs). All constructs were sequenced to confirm that mutations were correctly incorporated and are listed in Supplementary Tables 6C (assignment constructs) and 6D (other constructs).

### Protein expression

Plasmid was transformed into BL21(DE3) cells (StrataGene) and grown overnight at 37 °C in an LB pre-culture containing 50 µg/ml kanamycin and 34 µg/ml chloramphenicol. The content of the growth media depended on the labeling scheme, where adequate antibiotics were added in all cases:

1a. For methyl NMR-based experiments, non-labeled (NMR inactive) constructs were expressed in $D_2O$ based minimal medium, which contained ~95% $D_2O$ and ~5% $H_2O$ (referred to as r$D_2O$ M9 medium), supplemented with 0.5 g/L $^{14}NH_4Cl$ and 4 g/L $^1H^{12}C$ glucose.

1b. Ile-δ1[$^{13}CH_3$] and Met-ε1[$^{13}CH_3$] labeled (IM-labeled) proteins were expressed in 99.8% $D_2O$ based minimal medium (referred to as $D_2O$ M9), supplemented with 0.5 g/L $^{14}NH_4Cl$ and 2 g/L $^2H^{12}C$ glucose, except for monomer assignment mutants, which were expressed in r$D_2O$ M9 medium.

2a. For Csl4 backbone assignments, Csl4 was expressed in $H_2O$ based minimal medium supplemented with 0.5 g/L $^{15}NH_4Cl$ and 2 g/L $^1H^{13}C$ glucose.

2b. For Csl4 sidechain assignments, Csl4 was expressed in $D_2O$ based minimal medium supplemented with 0.5 g/L $^{15}NH_4Cl$ and 2 g/L $^2H^{13}C$ glucose.

3. Constructs used for cryo-EM, X-ray crystallography and $^{19}$F NMR experiments were expressed in LB medium.

Cells were inoculated from the pre-culture into $H_2O$ M9 (1. and 2.) or LB (3.) medium to an $OD_{600}$ of 0.1 and grown at 37 °C to an $OD_{600}$ of 0.6 – 0.8. For non-deuterated and LB media (2a. and 3.), cells were then induced with 1 mM isopropyl ß-D-1-thiogalactopyranoside (IPTG). For deuterated media (1. and 2b.), cells were spun down (15 min, 1300 × g) and resuspended into r$D_2O$ M9 (1a.) or $D_2O$ M9 (1b. and 2b.) medium, inoculated into another pre-culture to an $OD_{600}$ of 0.1 and grown overnight at 37 °C. On the next day, cells were diluted into fresh r$D_2O$ M9 (1a.) or $D_2O$ M9 medium (1b. and 2b.) to an $OD_{600}$ of ~0.2 and grown to an $OD_{600}$ of 0.6–0.8. At this point cells were either induced with 1 mM IPTG (1a.) or, if constructs were to be IM-labeled, 60 mg/L $^2H^{12}C$ (1b.) or $^2H^{13}C$ (2b.) ketobutyric acid-4-$^{13}CH_3$ and 100 mg/L L-

methionine-(methyl-$^{13}$C) (1b. and 2b.) were added and the cells were further incubated for 1 h at 37 °C prior to induction with 1 mM IPTG. Proteins were over-expressed for ~18 h at 20 °C, harvested by centrifugation (20 min, 6000 × $g$), after which cell pellets were stored at -20 °C until purification.

## Incorporation of tfmF using amber codon suppression

The non-natural amino acid 4-trifluoromethyl-L-phenylalanine (tfmF) was incorporated into Rrp41 using amber codon suppression. The *rrp41* gene including an N-terminal hexahistidine-tag and a TEV cleavage site was cloned into an ampicillin resistance pBAD vector by Gibson assembly[56]. An amber stop codon (TAG) was introduced at position G71, Q86 or D113 of Rrp41 using site-directed mutagenesis with primers as given in Supplementary Table 6B. The resulting plasmid was co-transformed with a tetracycline resistance pDule plasmid encoding for the tfmF amino-acyl-tRNA synthetase (tfmF-A65V-S158A) and its cognate suppressor tRNA$_{CUA}$[57] into Top10 cells (Thermo Fisher Scientific). The pDule-tfmF-A65V-S158A vector was a kind gift from Ryan Mehl (Addgene plasmid #85484; http://n2t.net/addgene:85484); RRID: Addgene_85484). An LB preculture containing 100 μg/ml ampicillin and 15 μg/ml tetracycline was grown overnight at 37 °C. Cells were inoculated into LB containing appropriate antibiotics to an OD$_{600}$ of 0.1 and grown to an OD$_{600}$ of 0.4 at 37 °C at which point 1 mM tfmF was added to the solution. Cells were further grown for 1 h at 37 °C, shifted to 20 °C and induced with 1% L-arabinose. Proteins were over-expressed for ~18 h at 20 °C, harvested by centrifugation (20 min, 6000 × $g$) after which the cell pellets were stored at -20 °C until purification.

## Protein purification

Cell pellets were resuspended in 50 mM sodium phosphate buffer, pH 7.4, 150 mM NaCl, 0.5 mM DTT, 10 mM imidazole (buffer A), except for Rrp44$^{D168N, D536N}$ where 500 mM NaCl was used. Next, 0.1% (v/v) triton X-100 and 1 mg/L lysozyme were added. Cells were subsequently lysed by sonication, cell debris was removed by centrifugation (30 min, 39,000 × $g$) and the filtered lysate (1.2 μm) was passed onto a gravity flow column filled with 4 ml Ni-NTA resin. The column was washed with 5–10 column volumes of buffer A and additionally washed with buffer A supplemented with 5 M NaCl for Rrp44$^{D168N, D536N}$ to remove RNA. Protein was eluted from the resin using buffer A supplemented with 300 mM imidazole. 0.5 mg TEV protease were added to the eluate that was dialyzed overnight into 20 mM HEPES, pH 7.5, 150 mM NaCl, 1 mM DTT (buffer B). All TEV-cleaved constructs contained Gly-Ala residues N-terminal to the protein sequence. The dialysate was subsequently passed onto a second Ni-column to remove the purification tag and TEV protease. The concentrated flow-through was subjected to a size-exclusion chromatography (SEC) purification step using a HiLoad 16/600 Superdex 200 pg (for full-length Rrp44) or a HiLoad 16/600 Superdex 75 pg (all other constructs) column in 10 mM HEPES buffer pH 7.5, 200 mM NaCl, 1 mM DTT (buffer C). For protein constructs that were subsequently linked to a TEMPO spin-label or reconstituted with a TEMPO spin-labeled construct (see below) a buffer devoid of DTT was used. Purity was assessed by sodium dodecyl sulfate poly-acrylamide gel electrophoresis (SDS-PAGE). Concentrations were determined by OD$_{280}$ measurements and extinction coefficients that were computed with ProtParam[58]. Purity of Rrp44$^{D168N, D536N}$ was further assessed by the OD$_{260}$/OD$_{280}$ absorption ratio, where only samples with an OD$_{260}$/OD$_{280}$ ratio below 0.75 were used.

## $^{19}$F-labeling with BTFA

wtRrp42, Rrp42$^{C59S}$ and Rrp42$^{C59S, A106C}$ were labeled with 3-Bromo-1,1,1-trifluoroacetone (BTFA) prior to SEC purification. To that end, 1 mM DTT was added to the concentrated protein (ca. ~200 μM, 2 ml) and the solution was incubated for 15 min at room temperature under gentle agitation. Subsequently, 20 mM BTFA was added after which the

linking reaction proceeded for 1 h at room temperature under gentle agitation. Excess label was removed in the final SEC purification step.

## TEMPO-labeling

Rrp42$^{C59S, A106C}$ and Csl4$^{C122S, E130C}$ were labeled with 4-maleimido-2,2,6,6-tetramethylpiperidine-1-oxyl (4-maleimido-TEMPO) prior to SEC purification. 4-maleimido-TEMPO was added in 3x excess to the DTT-free sample and incubated for 1 h at room temperature under gentle agitation. Excess label was removed in the final SEC purification step using DTT-free buffer C. No DTT was added to the sample when TEMPO spin-labeled protein was reconstituted into Exo9 or Exo10 (see below).

## Exosome reconstitution

To reconstitute Exo9 or Exo10, the individually purified subunits (or heterodimers) were mixed in stoichiometric amounts and incubated for 30 min at room temperature under gentle agitation. Next, the reconstituted exosome complex was purified by SEC with a HiLoad 16/600 Superdex 200 pg into buffer C (DTT-free if TEMPO spin-labeled protein was present). Successful reconstitution was assessed by SDS-PAGE (Supplementary Fig. 3). An uncropped SDS-PAGE gel is provided in the source data sheet and Supplementary Information page 59.

## X-ray crystallography

Crystals of Exo9 (8 mg/ml) grew in 0.2 M ammonium sulfate, 0.1 M sodium acetate pH 5.5, 10% PEG MME 2000, at 4 °C, by vapor diffusion. Diffraction data were collected at the PXII beamline of the Swiss Light Source (SLS, Villigen, Switzerland). The crystals diffracted to 3.8 Å resolution, belonged to space group $P2_12_12_1$ with cell dimensions of a = 100.525 Å, b = 148.383 Å, c = 195.065 Å, α = β = γ = 90°, and contained one molecule in the ASU. Data were processed and scaled with XDS (version XDS-INTEL64_Linux_x86_64_2022)[59]. The structure was solved by molecular replacement, using an in silico model (generated from the structures of the *S. cerevisiae* and *H. sapiens* homologs using MODELLER (version 10.1) and AlphaFold (accession codes AF-G0S755-F1, AF-G0SC21-F1, AF-G0S1P1-F1, AF-G0SCD1-F1, AF-G0RZG4-F1, AF-P0CT46-F1, AF-G0RZX8-F1, AF-G0S9A0-F1, and AF-G0SE33-F1 for Rrp45, Rrp41, Rrp43, Rrp46, Rrp42, Mtr3, Rrp40, Rrp4, and Csl4, respectively) as search model[13,60]. Iterative cycles of model building and refinement were carried out in COOT (version 0.9.6), PHENIX (version 1.20.1-4487) and ISOLDE (version 1.6)[61–63]. Data collection and refinement statistics are given in Supplementary Table 1A. Atomic coordinates have been deposited in the Protein Data Bank (PDB) with accession code 8PEL.

## Single particle cryo-EM

To increase the purity of the complex and to exchange to cryo-EM buffer, the Exo9 complex was subjected to a second SEC purification step using a Superdex 200 10/300 GL column in 20 mM sodium phosphate buffer pH 7.5, 100 mM NaCl (buffer D). The protein was diluted to a final concentration of ~2 μM. Quantifoil R1.2/1.3 Cu300 holey carbon grids were glow discharged twice for 100 s, at 15 mA and 0.39 mBar in an easyGlow system (PELCO). 3 μl of sample were applied to freshly glow discharged grids using a Vitrobot mark IV plunge freezer (ThermoFisher Scientific). After a 5 s incubation at 4 °C and 100% humidity, samples were blotted for 5 s using blot force 12 and plunged into liquid ethane. 6579 micrograph movies were collected on a CryoArm200 cryo-electron microscope (JEOL) equipped with a K2 direct electron detector (Gatan), in-column energy filter operated with slit width of 20 eV, and cold-field emission gun (low-flash interval 4 h). Data were recorded using SerialEM[64], in a 5 × 5 multi-hole pattern, in counting mode, with a total dose of 40 e$^-$/Å$^2$ fractionated over 40 frames, and defocus range from -0.6 to -2 μm.

The data processing pipeline is depicted in Supplementary Fig. 2. Data were processed using RELION (version 4.0.1)[65]. Particles were picked using the Topaz wrapper[66] within RELION and subjected

to multiple rounds of 2D and 3D classification to eliminate partially disassembled complexes. The final 3D reconstruction was obtained from 276,958 particles and refined to an overall resolution of 3.19 Å (FSC cut-off 0.143). Iterative cycles of model building and refinement, using the refined crystal structure as starting model, were carried out in COOT, PHENIX and ISOLDE[61–63]. Data collection and refinement statistics are given in Supplementary Table 1B. Atomic coordinates and density maps have been deposited in the PDB and in the Electron Microscopy Data Bank (EMDB) with accession codes 8R1O and EMD-1882, respectively.

## RNA in vitro transcription and purification

To ensure that the RNA was single-stranded, we designed RNAs consisting of random sequences of A and G nucleotides that are not expected to form any stable secondary structure elements. 80mer RNA was used for the activity assay to provide better readouts, whereas a shorter 46mer RNA was employed in the NMR experiments to avoid possible multimerization of the exosome complex. The sequences of the RNAs used in this study are listed in Supplementary Table 7. RNA was obtained by in vitro transcription using an in-house purified T7 polymerase containing a P266L mutation[67]. 1 μM template DNA and 1 μM T7 promoter oligonucleotide were mixed with 4 mM nucleotides, 30 mM MgCl$_2$, 10% (v/v) DMSO, 50 mM Tris pH 8.0, 0.01% (v/v) triton X-100, 1 mM spermidine, 5 mM DTT and 18 μg/ml T7 polymerase and incubated for 4 h at 37 °C. 50 mM EDTA at pH 8.0 was subsequently added to dissolve phosphates and the RNA was precipitated by adding 300 mM sodium acetate at pH 5.0 and 70 vol% isopropanol followed by incubation for at least 1 h at -20 °C.

RNA was purified by anion exchange chromatography using a preparative DNAPac 100 column (Dionex) at 60 °C with a linear buffer gradient from 20 mM Tris pH 8.0, 5 M urea (buffer E) to 20 mM Tris pH 8.0, 5 M urea, 1 M NaCl (buffer F). Fractions containing the desired RNA were pooled and RNA was precipitated by adding 300 mM sodium acetate at pH 5.0 and 70 vol% isopropanol and incubation for at least 1 h at -20 °C. Next, the RNA was pelleted, the supernatant was discarded and the pellet was washed with ethanol, dried at 37 °C for 3 h and re-suspended in H$_2$O. The purity of the preparation was assessed by Urea-PAGE on a 16% acrylamide gel.

## NMR sample preparation

Methyl-TROSY NMR experiments were performed in D$_2$O-based buffer C. The final sample, in H$_2$O-based buffer C, was first concentrated to 200 μl, diluted 75 times with D$_2$O-based buffer C and then concentrated again. For all other experiments, 10% of D$_2$O-based buffer C was added to the sample for frequency locking. 46mer RNA (see Supplementary Table 7) was added in 1.5 times excess to either Exo9 or Exo10 Rrp44$^{D168N, D536N}$, which is a mutant that inactivates both endo- and exonucleolytic activity of Rrp44. Protein concentrations varied depending on the construct: for exosome samples, concentrations were typically between 50 – 120 μM, while monomers or dimers could often be investigated at higher concentrations. For NMR experiments on the exosome complex, 200 μl sample was placed into an NMR tube with a diameter of 3 mm. For all other constructs 500 μl sample was placed into an NMR tube with a diameter of 5 mm. Constructs used in this study and experiments conducted on them are listed in Supplementary Table 6E.

## NMR spectrometers

NMR experiments were conducted on Bruker 500 MHz, 600 MHz and 800 MHz Avance Neo spectrometers (11.7 T, 14.1 T and 18.8 T magnetic field strength, respectively) equipped with triple resonance cryogenic TCI probeheads cooled with liquid helium (800 MHz) or liquid nitrogen (500 MHz and 600 MHz). For the 500 MHz and 600 MHz

spectrometers, the $^1$H channel was tuned to $^{19}$F frequency (471 MHz and 565 MHz, respectively) for fluorine NMR experiments. Experiments were acquired with Topspin 4.0.3 (500 MHz and 800 MHz spectrometer) or Topspin 4.0.8 (600 MHz spectrometer).

## Methyl NMR and backbone assignment experiments

2D methyl-TROSY spectra were collected using the SOFAST-HMQC pulse sequence[8] with carbon acquisition times of 30 ms (Exo9 and larger) or 60 ms (exosome monomers) and an interscan delay of 0.5 s at 313 K.

To assign Csl4 Ile-δ1 resonances, standard backbone assignment experiments (TROSY-based HNCACB, HNCA, HNCOCACB, HNCO, HNCACO) were conducted for the monomer and the Ile-δ1 methyl groups were assigned by standard TROSY-based H(CCCO)NH and C(CCO)NH experiments, in which either the Ile-δ1 $^1$H or $^{13}$C chemical shift is correlated with $^1$H$^N$ and $^{15}$N chemical shifts of the preceding residue. In total 137 out of 201 non-Pro residues (68 %, Supplementary Fig. 28, plotted with Sparky v3.115) and 10 out of 14 Ile-predecessors were assigned. Csl4 backbone assignments and $^1$H-$^{15}$N TROSY raw and processed data have been deposited in the Biological Magnetic Resonance Data Bank (BMRB) with BMRB ID 53248 [https://bmrb.io/data_library/summary/?bmrbId=53248]. Note, that assignments of Ile-δ1 via backbone assignments only complemented assignments that were obtained from Met and Ile point mutations (Supplementary Table 6C).

## $^{19}$F NMR experiments

Experiments were acquired with an acquisitions time of 0.05 s, 1–1.5 s interscan delay at 298 K using in-house developed pulse sequences[40,42,43]. Chemical exchange saturation transfer experiments (CEST) were conducted at $B_1$ field strengths as indicated in the figures (10–25 Hz) applied for $t_{CEST} = 400$ ms with 67 frequency offsets ranging from -2450 Hz to +2450 Hz. The central frequency (0 Hz) was set on resonance with the most intense fluorine resonance and frequency offsets were sampled symmetrically around 0 Hz. CEST intensities were referenced to the intensity determined for offsets at ±10,000 Hz. EXSY experiments were acquired using 12 mixing delays $t_{ZZ}$ (1, 2, 5, 10, 25, 50, 75, 100, 150, 200, 400, 600 ms) and an acquisition time in the indirect dimension of 14 ms. $R_1$ relaxation rates were determined employing an inversion recovery pulse sequence with an interscan delay of 2 s and with at least 9 delays $t_{R1}$ (for all samples without RNA and for Exo9 Rrp41$^{G71tfmF}$ Csl4$^{C122S, E130C-TEMPO}$ with RNA: 0.001, 0.05, 0.05, 0.1, 0.25, 0.5, 0.8, 1.5, 3, 5, 5, 8 s; for all other samples with RNA: 0.001, 0.05, 0.1, 0.25, 0.5, 0.8, 1.5, 3, 5 s). $R_1$ relaxation rates were obtained by fitting:

$$I(t_{R1}) = I_\infty \left(1 - 2\exp(-R_1 t_{R1})\right) \tag{1}$$

to experimental intensities $I(t_{R1})$ for varying delay times $t_{R1}$.

CEST and inversion recovery experiments were conducted on a 500 MHz spectrometer, EXSY experiments were conducted on a 600 MHz spectrometer.

Constant time Carr-Purcell-Meiboom-Gill (CPMG) relaxation dispersion (RD) experiments were conducted with a relaxation delay ($T_{CPMG}$) between 2 and 20 ms (see Supplementary Table 8) using at least 20 frequencies ($\nu_{CPMG}$) if $T_{CPMG} \geq 6$ ms and 10 frequencies if $T_{CPMG} < 6$ ms. The maximum frequency that was used was 5000 Hz and the minimum frequency depended on the length of $T_{CPMG}$ (see Supplementary Table 8). CPMG RD experiments were conducted on a 500 MHz spectrometer unless indicated otherwise.

## PRE experiments

For paramagnetic relaxation enhancement (PRE) experiments, an initial methyl-TROSY or $^{19}$F spectrum was acquired in the presence of non-reduced TEMPO spin-label providing intensities $I_{para}$. Then, 5 mM sodium ascorbate was added to reduce the spin-label and another

spectrum was acquired to obtain intensities $I_{dia}$. Methyl PREs were calculated as

$$\Gamma_{CH_3} = \frac{I_{para}}{I_{dia}} \qquad (2)$$

Additionally, for PRE experiments of tfmF-labeled samples, an initial $T_1$ inversion recovery experiment was acquired providing $R_{1,para}$ (Eq. 1). After addition of 5 mM sodium ascorbate another $T_1$ inversion recovery experiment was acquired to obtain $R_{1,dia}$ (Eq. 1). The $R_1$-based PRE, $\Gamma_1$, was calculated as

$$\Gamma_1 = R_{1,para} - R_{1,dia} \qquad (3)$$

### Data analysis

NMR data were processed using the NMRPipe/NMRDraw software suite (version 11.7)[68]. Methyl resonance intensities were obtained with NMRPipe while $^{19}$F resonance integrals were obtained using an in-house Matlab script. Assignments of Csl4 were performed in Cara (version 1.9.1.7)[69].

For model fitting, in-house Matlab scripts were employed. In all fitting routines, the target function

$$\chi^2 = \sum_{exp} \sum_{i=1} \left( \frac{O_{exp,i} - O_{calc,i}}{\sigma_{exp,i}} \right)^2 \qquad (4)$$

was minimized using the fminsearch routine in Matlab (version R2022b). In Eq. 4 $O_{exp,i}$ corresponds to an experimentally determined data point ($i$) in one of the recorded datasets $exp$ {$^{19}$F RD data of Exo9 in the absence of RNA at 471 and 565 MHz; CEST data in the absence of RNA at $B_1$ fields of 10 (Exo9 and Exo10), 15 (Exo10) and 25 Hz (Exo9 and Exo10); EXSY data of Exo10 in the absence of RNA, a 1D NMR spectrum of Exo10 in the absence of RNA; 1D NMR spectra Exo9 and Exo10 in the presence and absence of RNA and in the paramagnetic and diamagnetic states (to assess $\Gamma_2$); intensities in $R_1$ inversion recovery experiments for Exo9 and Exo10 in the presence and absence of RNA and in the paramagnetic and diamagnetic states (to assess $\Gamma_1$)}. $O_{calc,i}$ corresponds to a back-calculated value of an observable based on the model parameters, as described below. $\sigma_{exp,i}$ is an estimate of the measurement uncertainty for a data-point and is based on the noise level in the spectra or duplicate measurements.

A two-site exchange process between a ground state $G$ (the open conformation) and an excited state $E$ (the closed conformation) as described by the equilibrium $G \rightleftarrows E$, with $k_{ex} = k_{EG} + k_{GE}$, $p_E = \frac{k_{GE}}{k_{ex}}, p_G = \frac{k_{EG}}{k_{ex}}$, and $p_E = 1 - p_G$ was fitted to the data.

To reduce the number of fitting parameters and thus over-fitting of the data, we assumed that Rrp42$^{C59S, A106C-TFA}$ chemical shifts of the ground ($\omega_G$) and excited ($\omega_E$) states were the same for Exo9 and Exo10 complexes in the presence and absence of RNA. Furthermore, we assumed that the RNA bound complexes were 100% in the open conformation as demonstrated by PRE experiments.

The CPMG relaxation dispersion data was back-calculated numerically using the equations derived by Baldwin[70]. These equations provide an analytical solution of a system undergoing two-site exchange and are not limited to a specific timescale of the motion. $R_{2,inf}$ of the ground and excited states were assumed to be the same.

To fit the CEST data, the signal intensities ($I$) at offsets $\omega_{CEST}$ were back-calculated according to:

$$I(\omega_{CEST}) = I_{proj} * exp(Mt_{CEST})I_0 \qquad (5)$$

where $t_{CEST}$ is the time during which the weak $B_1$ field is applied. The equilibrium $z$ magnetization is $I_0 = (E/2 \quad I_x^G \quad I_y^G \quad I_z^G \quad I_x^E \quad I_y^E \quad I_z^E)^T = (1/2 \quad 0 \quad 0 \quad p_G \quad 0 \quad 0 \quad p_E)^T$, where $E$ is the identity operator and $I_{\{x,y,z\}}^{\{G,E\}}$ are the components of the magnetization vector in $x$, $y$, or $z$ direction for the $G$ and $E$ states, respectively. M is the evolution matrix (according to the Bloch-McConnell equations)[71]:

$$M = \begin{pmatrix} 0 & 0 & 0 & 0 & 0 & 0 & 0 \\ 0 & -R_2^G - k_{GE} & -\omega_G & \omega_1 & k_{EG} & 0 & 0 \\ 0 & \omega_G & -R_2^G - k_{GE} & 0 & 0 & k_{EG} & 0 \\ 2R_1^G p_G & -\omega_1 & 0 & -R_1^G - k_{GE} & 0 & 0 & k_{EG} \\ 0 & k_{GE} & 0 & 0 & -R_2^E - k_{EG} & -\omega_E & \omega_1 \\ 0 & 0 & k_{GE} & 0 & \omega_E & -R_2^E - k_{EG} & 0 \\ 2R_1^E p_E & 0 & 0 & k_{GE} & -\omega_1 & 0 & -R_1^E - k_{EG} \end{pmatrix}$$

where, $R_1^G$, $R_1^E$, $R_2^G$ and $R_2^E$ are the longitudinal and transverse relaxation rates of state $G$ and $E$, respectively. $\omega_G$ and $\omega_E$ (in rad/s) denote the offsets between the CEST frequency and the chemical shifts of the ground and excited states, respectively. $\omega_1 = \gamma_F B_1$ is the weak $B_1$ field (in rad/s) applied from the $y$-direction and $\gamma_F$ is the gyromagnetic ratio of fluorine. $I_{proj} = (0 \quad 0 \quad 0 \quad 1 \quad 0 \quad 0 \quad 1)$ is the vector that projects the magnetization onto $I_z^G + I_z^E$, which results in observable magnetization.

To fit the EXSY data, the experimental signal intensities were back-calculated according to[72]:

$$\begin{pmatrix} I_{GG} & I_{EG} \\ I_{GE} & I_{EE} \end{pmatrix} = S_{EXSY} * exp\left( \begin{pmatrix} -k_{GE} - R_1^G & k_{EG} \\ k_{GE} & -k_{EG} - R_1^E \end{pmatrix} t_{ZZ} \right) \begin{pmatrix} p_G & 0 \\ 0 & p_E \end{pmatrix} \qquad (6)$$

where $S_{EXSY}$ is a scaling factor of the experimental intensities, $I$ are the intensities of either the auto peaks of the ground ($I_{GG}$) or excited ($I_{EE}$) state, or the intensities of the cross peaks between the ground and excited states ($I_{EG}$ or $I_{GE}$) and $t_{ZZ}$ is the EXSY mixing time.

The 1D NMR spectra were simulated based on

$$I(\omega) = S_{spectrum} * \left| \Re \left( \Sigma \left( M^{-1} * I_0 \right) \right) \right| \qquad (7)$$

where $S_{spectrum}$ is a scaling factor for the intensity in a specific spectrum, $\omega$ is the offset, $I_0 = (p_G \quad p_E)^T$ and

$$M = \begin{pmatrix} -R_2^G + i(\omega_G - \omega) - k_{GE} & k_{EG} \\ k_{GE} & -R_2^E + i(\omega_E - \omega) - k_{EG} \end{pmatrix}$$ in the absence of PREs or

$$M = \begin{pmatrix} -R_2^G - \Gamma_2^G + i(\omega_G - \omega) - k_{GE} & k_{EG} \\ k_{GE} & -R_2^E - \Gamma_2^E + i(\omega_E - \omega) - k_{EG} \end{pmatrix}$$

in the presence of PREs, where $\Gamma_2^{\{G,E\}}$ are the $R_2$-based PRE effects in states $G$ or $E$.

To fit the $R_1$ PRE rates, the experimental signal intensities ($I$) from inversion recovery experiments were back-calculated according to[45]:

$$I(t_{R1}) = I_{proj} * exp(Mt_{R1})I_0 \qquad (8)$$

where $t_{R1}$ is the relaxation delay and $I_0 = (1/2 \quad p_G \quad p_E)^T$. The evolution matrix M is

$$M = \begin{pmatrix} 0 & 0 & 0 \\ 2\left(R_1^G + \Gamma_1^G\right)p_G & -R_1^G - \Gamma_1^G - k_{GE} & k_{EG} \\ 2\left(R_1^E + \Gamma_1^E\right)p_E & k_{GE} & -R_1^E - \Gamma_1^E - k_{EG} \end{pmatrix}$$

where the $\Gamma_1^{\{G,E\}}$ are the $R_1$-based PRE effects in states $G$ or $E$ and $I_{proj} = \begin{pmatrix} 0 & 1 & 1 \end{pmatrix}$ is the projection vector that results in observable magnetization.

The determined $\Gamma_1^G$, $\Gamma_1^E$, $\Gamma_2^G$ and $\Gamma_2^E$ values were used in combination with the Solomon-Bloembergen equations to extract order parameters $S^2$ of the ground and excited state according to[52,73,74]

$$\Gamma_1 = \frac{1}{r^6} \frac{2}{5} \left(\frac{\mu_0}{4\pi}\right)^2 \gamma_I^2 g^2 \mu_B^2 s(s+1) J(\omega_I)$$

$$\Gamma_2 = \frac{1}{r^6} \frac{1}{15} \left(\frac{\mu_0}{4\pi}\right)^2 \gamma_I^2 g^2 \mu_B^2 s(s+1)\left(4J(0) + 3J(\omega_I)\right) \quad (9)$$

where the spectral density $J$ is expressed as $J(\omega) = \frac{S^2 \tau_c}{1+(\omega\tau_c)^2} + \frac{(1-S^2)\tau_t}{1+(\omega\tau_t)^2}$, $\tau_c^{-1} = \tau_R^{-1} + \tau_s^{-1}$ and $\tau_t^{-1} = \tau_r^{-1} + \tau_s^{-1} + \tau_i^{-1}$ [52,75]. In these equations $r$ is the distance between the spin-label and the probing nucleus (in the ground or excited state), $\mu_O$ is the permeability of vacuum ($1.257 \times 10^{-6}$ N A$^{-2}$), $\gamma_I$ is the gyromagnetic ratio of flourine ($251.815 \times 10^6$ rad T$^{-1}$ s$^{-1}$), $g$ is the Landé factor (-2.002), $\mu_B$ is the magnetic moment of the free electron (-$9.285 \times 10^{-24}$ J T$^{-1}$), $s$ is the electron spin quantum number ($\frac{1}{2}$), $\omega_I$ is the Larmor frequency of a flourine nucleus (471 MHz), $S^2$ is the squared order parameter of the ground or excited state and $\tau_i$ is the correlation time of the vector connecting the spin-label and the probing nucleus in the ground or excited state, $\tau_R$ is the rotational correlation time of the protein complex (assumed to be 100 ns for Exo9 and 140 ns for Exo10) and $\tau_s$ is the electron relaxation time (assumed to be 100 ns). The lower sensitivity of $\Gamma_2$ as compared to $\Gamma_1$ for fast internal motions follows from Eq. 9, in which the term $J(0)$ dominates the value of $\Gamma_2$. $J(0)$ in turn is dominated by motions of the entire complex ($\tau_R$) as long as $S^2$ is not too small. On the other hand, $\Gamma_1$ depends only on $J(\omega_I)$, which is sensitive to fast internal motions ($\tau_i$).

The 1D $^{19}$F spectrum and $T_1$ data of diamagnetic Exo10 Rrp42$^{C59S-A108C-TEMPO}$ Rrp41$^{D113tfmF}$ were deconvoluted by fitting the 1D $^{19}$F spectrum to a double Lorentzian function. The thus obtained peak positions and line widths were fixed for the deconvolution of $T_1$ data, for which only intensities of both peaks were parameters. Equation 1 was fitted to those intensities to obtain $R_1$ relaxation rates for the deconvoluted resonances.

Uncertainties in the fitted model parameter (which are globally: $p_E$ in Exo9 and Exo10, $k_{ex}$ in Exo9 and Exo10; for Rrp42$^{C59S, A106C-TFA}$: the chemical shifts of the ground and excited states, $R_{2,inf}$ at 471 MHz and 565 MHz, $R_1$ and $R_2$ in Exo9 and Exo10, scaling factors for EXSY spectra and for PRE spectra in Exo9 and Exo10 in the absence and presence of RNA; for Rrp41$^{D113tmF}$: the chemicals shifts of the ground and exited states in Exo9 and Exo10, $R_1$ and $R_2$ in Exo9 and Exo10 in the absence and presence of RNA, $\Gamma_1$ and $\Gamma_2$ for Exo9 and Exo10 for the ground and excited states and scaling factors for the PRE $R_1$ inversion recovery experiments; for the Solomon-Bloembergen equations: $S^2$, $\tau_i$ and $r$ for the ground and excited states) were obtained from Monte-Carlo simulations where 200 artificial datasets were created based on the measurement uncertainties. Subsequently, the model was fitted to these datasets. In that procedure the starting parameters for the fit were varied randomly by 5% to prevent model bias. Model parameters are reported as best fit value +/- one standard deviation. The distributions of the fitting parameters, which are not necessarily Gaussian, are displayed in Supplementary Fig. 20.

## Chemical shift perturbations

Chemical shift perturbations (CSPs) in $^1$H ($\Delta\delta_H$) and $^{13}$C ($\Delta\delta_C$) dimensions were combined to a global CSP ($\Delta\delta$) by:

$$\Delta\delta = \sqrt{\left(\left(\frac{\Delta\delta_C}{4}\right)^2 + \Delta\delta_H^2\right)} \quad (10)$$

## Molecular Modeling

To generate a complete model of the Exo9 complex, we predicted the missing loops of the here obtained cryo-EM structure using AlphaFold2-based[13] ab intio structure prediction. Employing a local implementation of ColabFold 1.5.3[76] we modeled five structures of the complete Exo9 complex, each of which was refined in 12 iterative refinement cycles. The models were ranked based on the Local Distance Difference Test (lDDT)[77] and the Template Modeling score (TM)[78]. Subsequently, the models were optimized with the original united atom AMBER force field[79]. To complete the experimentally unresolved missing loops, we locally aligned the predicted structure to the cryo-EM structure and added the missing loops from the predicted structure to the experimental structure.

The resulting completed structure represents the open state of the Rrp42-EL (residues 77 to 117) assigned based on the NMR results (Fig. 4, and Supplementary Fig. 18) as the distance between Asp113 of Rrp41 and Ala106 of Rrp42 is with 51.3 Å larger compared to the experimentally expected distance of $6.8 \pm 0.1$ Å for the closed state (Supplementary Table 4). To obtain a structure of the closed state, Rrp42-EL was manually modeled into the unoccupied cavity near Rrp41 to achieve a short spin label distance comparable to the one observed in the $^{19}$F PRE experiments (Fig. 4, and Supplementary Fig. 18). During the manual modeling we iteratively repeated the process of interactively updating the coordinates of Rrp42-EL and subsequently energy optimized the loop using the MAXIMOBY program suit version 2023 (CHEOPS, Germany) with the original united atom AMBER force field[79]. The resulting structure reflects the closed state with a distance of 14 Å between Asp113 of Rrp41 and Ala106 of Rrp42. The structures of both states were further refined to the cryo-EM density map with molecular dynamics flexible fitting (MDFF)[80,81]. MDFF runs were set up with QwikMD[82] and performed with NAMD 2.13[83] employing the CHARM36 force field[84]. During an initial 800 step minimization phase, existing secondary structure elements (α-helix, β-sheet) as well as peptide isomerism (cis/trans) and center chirality were conserved. This was followed up by a 40 ns simulation phase at 300 K in implicit solvent. The refined structures were energy optimized employing MAXIMOBY (CHEOPS, Germany). The resulting structures are available at https://doi.org/10.5283/EPUB.77450.

## MD simulations

To adapt the structural models to aqueous environment and study the loop dynamics in solution at room temperature (293.15 K) we ran MD simulations initiated by the two completed Exo9 models of the open and closed state obtained as described above. The structures were protonated based on the local pKa values of each residue calculated at a pH value of 7 following Nielsen and Vriend[85]. Water molecules of the first and second solvation shell of the protein complex were set using a Vedani-like algorithm[86] implemented in MAXIMOBY. To prevent self-interactions of the protein due to periodic boundary conditions within the MD simulation, a cubic simulation box with the dimensions $17.97 \times 17.97 \times 17.97$ nm was set and filled with water molecules, sodium and chloride ions, at physiological conditions using the solvation workflow implemented in GROMACS 2021[87]. Steric clashes between the hydrogen of the solvation shell and bulk water were locally resolved through energy optimization in MAXIMOBY (CHEOPS, Germany). Subsequently we performed MD simulations with GROMACS 2021[87]

utilizing the OPLS/all-atom force field[88]. First, the system was heated to room temperature (293.15 K) in 1 ns with a step size of 1 fs within a nVT simulation, meaning the number of atoms ($n$), the volume ($V$), and the temperature ($T$) is constant, while the pressure ($p$) is flexible. The temperature was kept constant using a V-rescale thermostat[89] with a coupling constant 0.1 ps. The heating was performed in two steps, over the first 100 ps the temperature was raised continuously from 0 K to 100 K, in the following 900 ps the temperature was raised continuously to 293.15 K. The heating procedure was followed by a 1 ns nVT simulation with a stepsize of 1 fs under the same conditions as for the heating but at 293.15 K. Next, we performed a 10 ns npT run with a step size of 1 fs in which the number of atoms, pressure and temperature remained constant using a Berendsen barostat[90] (coupling constant 0.5 ps) and a V-rescale thermostat[89] (coupling constant 0.1 ps), while the volume was kept flexible. Following these equilibration steps we performed a 100 ns npT production run (step size 2 fs) for each of the two systems that was used for further analysis. In the production run, the temperature was controlled by a Nosé-Hoover thermostat[91,92] (coupling constant 0.5 ps) and a Parrinello-Rahman barostat[93] (coupling constant 2.5 ps) which perform better for equilibrated systems. All run parameter mdp files are available at https://doi.org/10.5283/EPUB.77450.

### Simulation evaluation

To evaluate the stability of the simulation we calculated the root mean square deviation (RMSD) of the $C^\alpha$ atoms of each snapshot of the simulation trajectory compared to the starting structure. Changes in the secondary structure were monitored using the defined secondary structure of proteins (DSSP) algorithm[94]. Inter and intra protein sub-unit interactions were determined with the contact matrix algorithm in MAXIMOBY (CHEOPS, Germany) and the PyContact plugin[95] for VMD 1.9.4. and PyMOL 3.0 were used for visual inspection of protein structures and simulation trajectories.

To obtain a representative structures of the MD simulations, each 0.1 ns frame of the simulation was analyzed for the current conformation and contacts of each residue within the system, resulting in a contact matrix for every frame. Each contact and conformation were weighted based on their importance. A mean matrix across the second half of the simulation was calculated and scored against every frame of the simulation. The frame most closely resembling the weighted mean vector was defined as representative for the simulation. The resulting representative MD simulations structures are available at https://doi.org/10.5283/EPUB.77450.

The number of interactions in Fig. 5C was obtained by statistical analysis of the simulation based on the contact matrix algorithm implemented in the MAXIMOBY program package (CHEOPS, Germany). Each contact with a percentage presence of over 40% across the second half of the simulation was deemed significant for the count.

For the comparison of distances in our MD simulations with derived distances from NMR experiments, we needed an estimate for the labels (TEMPO spin labels and tfmF) and their flexibility as they are not incorporated in our MD simulations. The approximation for the distance between the spin label and tfmF were considered based on our static structural models with incorporated labels within the Rrp42-EL open and closed state (Supplementary Fig. 22). We measured the maximum distance of the $C^\alpha$ atom and the corresponding label (carbon of the $CF_3$ group in tfmF; nitrogen of the nitroxide in TEMPO) resulting in a distance of 7 Å for tfmF and 6 Å for TEMPO. We consider this as the upper limit of the conformation space covered by the labels and added this as a gray bar to Supplementary Fig. 22 to reflect the actual experimentally measured possible distance range between the labels.

### Activity assays

**Urea-PAGE analysis.** 5 μM 80mer RNA (see Supplementary Table 7) was mixed to 10 mM HEPES buffer pH 7.5, 200 mM NaCl and 5 mM MgCl₂ and incubated for 5 min at 40 °C, after which a reference sample (0 min) in absence of the exosome was taken. Next, 1 μM exosome was rapidly mixed with the RNA-buffer solution and samples were taken at different time points (1, 2, 4, 8, 16, 32, 64, and 128 min) while the reaction proceeded at 40 °C. For each sample, the reaction was stopped by rapidly mixing 2x Urea-PAGE loading dye containing 8 M Urea, 20 mM EDTA, 2 mM Tris-HCl pH 8 and 0.0001% (w/v) bromophenol blue in a 1:1 ratio. The activity was qualitatively assayed by Urea-PAGE on 16% acrylamide gels. Uncropped Urea-PAGE gels are provided in the source data sheet and Supplementary Information page 60.

**HPLC analysis.** 5 μM 80mer RNA (see Supplementary Table 7) was mixed to 10 mM HEPES buffer pH 7.5, 200 mM NaCl and 5 mM MgCl₂ and incubated for 5 min at 40 °C, after which a reference sample (0 min) in absence of the exosome was taken. Next, 0.5 μM exosome was rapidly mixed with the RNA-buffer solution and samples were taken at different time points (1, 2, 3, 5, and 10 min) while the reaction proceeded at 40 °C. For each sample, the reaction was stopped by rapidly mixing a three times volume excess of 8 M Urea and heating the sample to 95 °C. Next, the nucleotide and RNA concentrations of each sample were determined by high-performance liquid chromatography (HPLC) using anion exchange on an analytical DNAPac PA100 column (Thermofisher) heated to 40 °C. The samples were applied onto the column using buffer E supplemented with 100 mM NaCl and eluted using gradient steps of buffer F (described in Supplementary Table 9). Elution peaks of the nucleotides and the RNA were integrated and concentrations were obtained by comparing integrals to a calibration curve for GMP with known concentrations. The concentrations were scaled by the ratio of the extinction coefficient of the RNA (for the RNA elution peak) or an average extinction coefficient per nucleotide (for the nucleotide elution peak) divided by the extinction coefficient of GMP. The experiments were conducted for three distinct protein batches that were independently expressed and purified, with three technical repeats each.

To obtain the catalytic rate $k_{cat}$, a linear equation, for which the slope is $k_{cat}$, was simultaneously fitted to the linear regime (time points 0 – 3 min for wtExo10 and Exo10 Rrp42$^{\Delta93-125}$, time points 0–10 min for Exo10 Rrp45-L and Exo10 Rrp45-L Rrp42$^{\Delta93-125}$) for the nucleotide and RNA data. Since it is not straightforward to identify the linear activity regime, we also analyzed the data for alternative linear activity regimes by including/excluding further data points in the linear fit as outlined in Supplementary Table 10A and shown in Fig. 6 and Supplementary Fig. 26. Catalytic rates are shown in Supplementary Table 10B. Importantly, irrespective of the data points analyzed our conclusion, namely that the catalytic rates of wtExo10 and Exo10 Rrp42$^{\Delta93-125}$ are identical within error limits, while Exo10 Rrp45-L Rrp42$^{\Delta93-125}$ shows enhanced activity compared to Exo10 Rrp45-L, holds.

### Sequence alignments

The sequences of the subunits ctRrp41, ctRrp42 and ctRrp45 were aligned with homologous protein sequences from human, *S. cerevisiae* and the *S. solfataricus* using Clustal Omega[96].

### Reporting summary

Further information on research design is available in the Nature Portfolio Reporting Summary linked to this article.

## Data availability

All data have either been deposited in public repositories or are available in the manuscript, the supplementary information or source data sheet. Atomic coordinates have been deposited in the Protein Data Bank (PDB) with accession code 8PEL (crystal structure) and 8R1O (cryo-EM structure). Cryo-EM maps have been deposited in the Electron Microscopy Data Bank (EMDB) with accession code EMD-18825. Csl4 backbone assignments and ¹H-¹⁵N TROSY raw and processed data have been deposited in the Biological Magnetic Resonance Data Bank (BMRB) with

BMRB ID 53248. All other raw and processed NMR data have been deposited in BMRBig with entry ID BMRbig108 (https://bmrbig.org/released/bmrbig108). The representative output structures of the MD simulations, the initial structures of the MD simulations and all run parameters files are available at (https://doi.org/10.5283/EPUB.77450). Source Data are provided as a Source Data file. Source data are provided with this paper.

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

## Acknowledgements

We thank Iris Holdermann (MPI Tübingen) for support in the crystallization of the Exo9 complex, Janina Petters for help with the cloning, Nadine Stefan and Johanna Stöfl for excellent technical assistance, Jan Overbeck and Philip Wurm for support in conducting NMR experiments and David Stelzig for assistance with the RNA activity assays. We thank Martine Sprangers from ("https://sprangers-fotografie.nl/") for converting the supplementary movie to generally readable format. All present and past group members are acknowledged for critically discussing the results during the course of the project. This project is funded from the European Union's Horizon 2020 research and innovation program under the Marie Skłodowska-Curie grant agreement No. 89550 (to JL), by the German Research Foundation (Deutsche Forschungsgemeinschaft) under grant agreement No. SP 1324/3-1 (to RS) and by the European Research Council under the European Union's Seventh Framework Program (FP7/2007–2013), ERC grant agreement No. 616052 (to RS).

## Author contributions

Conceptualization: R.S., J.L., D.L., and T.R. Methodology: R.S., J.L., D.L., and T.R. Investigation: J.L., D.L., A.B., M.P., and T.F. Formal Analysis: R.S., J.L., D.L., T.R., and T.F. Visualization: J.L., D.L., and T.F. Funding acquisition: R.S., J.L., and T.R. Project administration: R.S. Supervision: R.S. and T.R. Writing – original draft: R.S., J.L., D.L., T.R., and T.F. with consent of all the authors. Writing – review & editing: R.S., J.L., D.L., T.R., and T.F. with consent of all the authors.

## Funding

## Competing interests

The authors declare no competing interests.
