## [Transparent Peer Review file · Nature Communications]

4D structural biology: quantitative dynamics in the eukaryotic RNA exosome complex

Corresponding Author: Professor Remco Sprangers

Version 0:

Reviewer comments:

Reviewer #1

(Remarks to the Author)

The manuscript "4D structural biology: quantitative dynamics in the eukaryotic RNA exosome complex" by J. Liebau et al. describes how combining NMR experiments with X-ray structures, cryo-EM structures, assays and MD simulations can altogether provide new insight into dynamics of large biomolecular assemblies (~ 500 kDa). The authors focus on the RNA exosome complex, a 3' to 5' RNA exonuclease. They report the structure of this complex from eukaryotic thermophile *Chaetomium thermophilum*, and conclude that the architecture of this complex is identical to the structures of human and yeast RNA exosome complexes. The authors then point out a known issue of structure determination techniques: structures of flexible loops are often missing, and these flexible parts may be important for the complex function. The authors suspect and then demonstrate with their studies that the extended loop region in Rrp42 protein is one such loop that is important for controlling the RNA pathway inside the exosome. They demonstrate that one loop in Rrp42 protein inhibits the entry of RNA directly into Rrp44 exonuclease site. Because of this loop, RNA enters this Rrp44 site only through the large barrel-like body of the exosome. The experiments are extensive and the results seem to be supported by the performed study. The identification of the above-mentioned extended loop as controlling the RNA pathway is a significant result. I recommend the paper for publication, after the authors consider the comments below.

1) Figure S21 shows that RMSDs of open and closed Rrp42-EL (extended loop) both remain at < 4 Angstrom within 100 ns. The authors refer to this figure as showing the structures to be energetically stable. However, the RMSDs of the same two loops shown in Fig 5B are very different. Can the authors clarify the discrepancy?

2) The results indicate that Exo9 has the Rrp42-EL 95% in the open state, whereas the same loop in Exo10 is 74% of time/population in the open state. The paper states that the loop closed state is presumed to be blocking the aberrant RNA pathway. Figure 6 confirms that kinetics of RNA degradation in Exo10 is different in Exo 10 complexes with and without the loop. How do the authors explain that the loop being only 26% of time in the closed state (or that only 26% of Exo10 complexes have closed loop when RNA is entering) is sufficient to block the aberrant RNA pathway? Is it reasonable to call the loop "flexible plug" (abstract) given the 26% closed state population?

3) At the end of page 7, it is stated that the closed state was obtained by interactively modeling Rrp42-EL into the unoccupied cavity near Rrp41. It would be good to explain in more detail what is meant by "unoccupied cavity" (density in cryo-EM or physical cavity?). The figures show open space rather than a cavity where the loop is present. The SI description of modeling (pg S13) then mentions that Asp113/Rrp41 and Ala106/Rrp42 is experimentally expected to be 7 Angstrom, and the structure resulting from modeling has this distance at 14 Angstrom. What could be the error/deviation in the experimental result (7 Angstrom plus/minus what?), and what is the distribution of the specified distance in MD simulations (in the closed state)? Can the authors provide further discussion reflecting their confidence in the modeled structure truly representing the closed structure in experimental systems?

Reviewer #2

(Remarks to the Author)

The article entitled "4D structural biology: quantitative dynamics in the eukaryotic RNA exosome complex" by Jobst et al. have combined crystallography, cryo-EM, Methyl-TROSY NMR, 19F-NMR, and MD simulation to understand the structural and dynamic mechanism of RNA exosome from the eukaryotic thermophile *Chaetomium thermophilum* (ctExo9). Current

structures of exosome represent some snapshots of exosome function cycle. However, the extended loop region in Rrp42 (Rrp42-EL) and the entry loop in Rrp41 are largely unresolved in the current structures and it's unclear how the movement of these loop regions related to the exosome function. To address this fundamental and challenging question, the authors have used several different biophysical tools. Particularly, 19F-NMR revealed that Rrp42-EL samples closed and open conformations in both Exo9 and Exo10 complexes in the absence of RNA, while only the open conformation is sampled in the presence of RNA. Importantly, the authors also showed that the Rrp42-EL functions as a barrier to block an aberrant access path to the Rrp44 active site. Overall, the experiments are of high quality and the results are reasonably interpolated and the conclusion is clear. This is a great example how a collection of biophysical methods could complement each other to achieve deeper mechanistic understandings. However, I do have some questions and suggestions that may improve the manuscript.

1. In the "Main Text", the introduction may need to be improved a little bit. I didn't get what's the authors initial motivation for the work and what particular questions the work is going to answer. There are Exo9 and Exo10 structures from yeast and human. Why the authors want to use crystallization and cryo-EM to get the structure of Exo9 from an eukaryotic thermophile? I think expanding the introduction would make readers better understand the work.
2. In the section "RNA displaces a channel exit loop", the authors described methyl TROSY spectra first, but the later referred figures, Fig. 4B, C, F, G, are 19F-NMR, which should be the content of the next section "Rrp44 and RNA modulate the Rrp42-EL dynamics". I guess the authors have made something wrong there. I guess the referred figures might be Fig. 3x? Please make it clear.
3. The complex has ribonucleolytic activity, how can the NMR spectra be recorded in the presence of RNA?
4. In Fig S15, could you explain what the right panels of the CEST profiles are? In Fig S15A, the peaks of the two conformations coalesced together, for which the CEST should not work. Why the authors say it identified a minor state in Line 165?
5. I am a little concerned about the interpolation of the R1 PRE effect of Exo10-Rrp41D113tfmF in the absence of RNA (Figure 4F). The protein should sample multiple conformations (at least 2) as is proven in Exo9 and also in A106C-BTFA labeling. While the 19F spectra showed a clear coalesced peak as Fig 4c for Exo9, the 1D-19F spectra of Exo10-Rrp41D113tfmF without RNA look like a partial overlapping of a broad peak with a sharp peak. The authors seem plotting the signal as single peak for characterizing the R1 PRE effect. Of note, the observed R1 PRE effect will be affected by the exchange rate and also the population of the two states (one features strong PRE, while another features weak PRE) as discussed in Nat Chem Biol 16, 1006–1012 (2020). <https://doi.org/10.1038/s41589-020-0561-6>. In the case of Exo9-No RNA, the conformation exchange rates are much faster than the R1 PRE, which should result in an averaged observed R1. That means, the real R1 PRE for the closed conformation will be dramatically larger, due to only 5% of closed conformation. In the case of Exo10-No RNA, the exchange rate is slowed down, but may still significantly large than the R1 PRE. I would suggest the authors try to deconvolute the spectra to see are the two peaks relax similarly. Lower the temperature may also help to shift the exchange rate and thus the R1 relaxation behaviors of the two peaks. In the manuscript, the authors didn't consider such effect on the R1 PRE. Therefore, I would suggest to including this into the R1 PRE data analysis and discussion.
6. The authors may need to give references about how the order parameters are simulated.

Reviewer #3

(Remarks to the Author)

Recommendation:

Accept with minor corrections

Liebau et al. present an excellent study highlighting the complementary nature of dynamic NMR experiments and MD simulation to static X-ray or cryo-EM structures, focusing on a eukaryotic RNA-degrading large complex as their representative target. Well-executed methyl and fluorine NMR experiments allow an in-depth investigation of various interacting subunit loops and their role in facilitating or preventing RNA exit and entry into the exosome. Their conclusions are clearly drawn from the presented evidence without far-reaching extrapolation with clear and intricate figures, with minor flaws in methodology and data presentation. The authors performed an immense amount of design, purification, and characterization work that is to be commended. The reviewers recommend the manuscript for publication with the following suggested minor changes for improvement.

Comments:

The reviewers recommend submitting all raw and processed NMR data to a database such as BMRBig (<https://bmrbig.org>) to facilitate open science practices.

The reviewers recommend changing the reference citation for the 2D methyl-TROSY via SOFAST-HMQC to the following, as the current reference, while utilizing a similar pulse program, is not the correct one:

o Amero et al. Fast two-dimensional NMR spectroscopy of high molecular weight protein assemblies. 2009. J. Am. Chem. Soc., 131 (10), 3448-3449. <https://pubs.acs.org/doi/10.1021/ja809880p>

The video file (Movie S1) associated with this manuscript submission could not be opened for evaluation. Please check compatibility with all OS.

The reviewers recommend bringing Supp Fig S26 Panel C to the main text and combining with Fig 6. In that same idea, we

recommend removing Fig S26D (repetitive with Fig 6B), and potentially consider combining Fig S25, Fig S26A-C into Fig 6 if space allows.

Please add how you determined the linear activity regime in the Supplemental Methods section near lines 428-431. The explanation is somewhat buried in the text in the caption for Figure S26. Visually, the excluded points look linear, and so further clarification of how you excluded those points is needed.

The reviewers recommend clarifying the protein sequences used for CS1, CS2, and PIN constructs, as only the Rrp44 is denoted in the UniProt table (Supp Table S5).

Please clarify what peaks on the SEC correspond to the pooled fractions for Exo9 & Exo10. There are 3 peaks evident in the chromatogram, one of which overlaps in both samples, but two that are shifted between Exo9 and Exo10. (Supp Fig S3B).

The gel in Supp Fig S3C has additional, higher molecular weight, unlabeled bands between Rrp42 and Rrp44. Please clarify if you think these are oligomer bands of the subunits or possible degradation of the exosome, or a contaminant from the Rrp44 purification.

The main text mentions Rrp6 (line 59-60) but this protein is not included in the study, and does not clearly connect the preceding and proceeding sentences. The reviewers recommend removing the sentence to improve clarity and manage reader expectations.

The conclusion of the text feels almost separate from the results presented. The reviewers recommend, between the two sentences on lines 287-292, to quickly summarize the “quantitative and functionally important information” in the “invisible structures” gleaned from the exosome that are presented in the text.

Parallel to the previous comment, the introduction lacks the central question or thesis statement you are trying to answer. It is contained within the abstract (“Our work thus demonstrates...thereby adding the time domain into structural biology.”). The reviewers recommend defining this 4th dimension aspect in the main text first paragraph, either in conjunction with or after the sentence “This thus opens ample opportunities...or in silico tools” to clearly connect the title, abstract, and project rationale.

It is stated in the main text that “the spin-label gives rise to enhanced R1 and R2 relaxation rates...” (lines 119-121). However, Fig S8 shows only the PRE (T1) values, which are within standard deviation from each other (i.e. no statistically significant enhancement). Table S3A also shows only the R1/T1 values. Please include the R2 rates (if calculated) or adjust the text to reflect the qualitative data in Fig S8B.

The sentence in lines 209-211 are confusing. Please rewrite or split the ideas to improve clarity, for example: “Our data thus imply that the invisible Rrp42-EL is more rigid in the closed conformation sampled more prominently by the Exo10 complex than in the ensemble of open conformations populated by the Exo9 complex.”

Please clarify if the RNA sequence is crucial for Exo10 function and the selection criteria of the 46-mer/80-mer comprised of just A/G bases used in the study (Supp Table S7).

It is stated in the main text that the Exo10 complex with Rrp42C59S, A106C-TFA retains ribonucleolytic activity, which is true, however the gel shows an enhanced activity compared to the WT (Fig S14) in contrast to the Fig S14 caption that states there is a “highly similar rate.” Please indicate why you think this rate appears increased compared to non-modified Rrp42.

Please clarify why the Rrp44 band so much stronger than the other subunits in Exo10 (Fig S3C) if the protein addition was stoichiometric (Supp Methods; lines 123-128) Additionally, please address why the band intensity of shared subunits between Exo9 and Exo10 is vastly different (Fig S3C).

Figure edits:

Major

Fig 3 panels are mislabeled in the main text as Fig 4 panels (section “RNA displaces a channel exit loop” from lines 152-155).

Minor

Fig 1: The label “Rrp44/Dis3” is confusing, as Dis3 is the human name and is never utilized nor mentioned again. The reviewers recommend removing “Dis3” from the figure and just denoting the subunit complex as Rrp44.

Fig S7: Please enlarge the spectra.

Fig S9: Please enlarge the graphs.

Fig S10 (line 138 in main text): Please change the cyan to another color for easier reading (e.g. dark blue, purple).

Fig S11A (line 144): Fig S11 does not contain panel A, correct in text.

Fig S12: The reviewers recommend putting small legend of orange = Ipara and black = Idia on the graph rather than in the caption, like in Fig S8A. Please enlarge the spectra.

Fig S16: Please change the cyan to another color (darker). Please enlarge the figure.

Reviewer #4

(Remarks to the Author)

In the work by Liebau et al., the authors used sophisticated NMR spectroscopy to analyze the structural dynamics of eukaryotic RNA exosome complex. Using the eukaryotic thermophile *Chaetomium thermophilum* exosome complex as a working model, the authors proved the methyl-group and fluorine NMR experiments can detect flexible or dynamic structural elements that are unavailable by crystallography or cryo-EM. Their NMR spectroscopy results agree with the previous determined structural studies of the eukaryotic RNA exosome complexes. But they identified an extra-loop of Rrp42 that appears to be a plug to block an aberrant route of RNA towards the active site of the exosome. This work is largely a demonstration of the state of the art of NMR spectroscopy on multi-subunit large complex by a tour-de-force work. The biological discovery is rather limited. Some specific questions for the authors to improve their work.

1. Previous works have shown that the Rrp44 interacts with Exo9 in at least two distinct conformations in the presence of various length of RNA substrates (for instance, 24 nt vs 12 nt). Have the authors studied the Rrp44's conformation change upon RNA binding and the interaction variation of Rrp41, Rrp45, Rrp42 with Rrp44 in the different RNA substrates?
2. Have the authors measured the PRE effects of Rrp42-EL with IM-labeled Rrp44 and the effects in the presence of RNA substrates?
3. The Rrp42 EL is not conserved among human, yeast and the thermophile exosome. Therefore, its potential role during RNA processing does not seem conserved. This reduces the biological relevance of the discovery on ctRrp42 EL. What could be the unique biological function of the ctRrp42 EL in a thermophile species?
4. The authors showed that Rrp45-L reduced the activity of the exosome, and channel-blocked Exo10 Rrp45-L with the Rrp42 Δ 93-125 recovered the exosome activity. This may imply the central channel partly recovered without Rrp42-EL plug, which does not necessarily prove the barrier block model of the direct access route.

Reviewer #5

(Remarks to the Author)

Version 1:

Reviewer comments:

Reviewer #1

(Remarks to the Author)

The authors addressed all the questions and issues and made the satisfactory corrections. I recommend the manuscript for publication.

Reviewer #2

(Remarks to the Author)

The revised manuscript addressed my concerns and I would recommend to publish on Nat. Comm. I have only one suggestion. It would be better to add in Methods or figure legends how the error bars are generated.

Reviewer #3

(Remarks to the Author)

All the points raised by the reviewer have been correctly addressed. I recommend publication of the article without further modification.

Reviewer #4

(Remarks to the Author)

The authors responded to my comments adequately. As stated in my comments in the first round of review, this work is technically sounding but does not provide much biological insights, which the authors claimed to pursue in their future studies. I do not have further comments on this work.

Reviewer #5

(Remarks to the Author)

We thank the reviewers for their time and efforts as well as for their positive responses regarding our manuscript. We apologize for our late response, which is due to illness of the last author. To ensure that future questions can be addressed quickly and adequately we added the contact information of the expert authors to the cover page of the manuscript.

Please find below our point-to-point response to the reviewer comments (that we highlighted with a bold/italic font).

REVIEWER COMMENTS

Reviewer #1 (Remarks to the Author):

*The manuscript “4D structural biology: quantitative dynamics in the eukaryotic RNA exosome complex” by J. Liebau et al. describes how combining NMR experiments with X-ray structures, cryo-EM structures, assays and MD simulations can altogether provide new insight into dynamics of large biomolecular assemblies (~ 500 kDa). The authors focus on the RNA exosome complex, a 3' to 5' RNA exonuclease. They report the structure of this complex from eukaryotic thermophile *Chaetomium thermophilum*, and conclude that the architecture of this complex is identical to the structures of human and yeast RNA exosome complexes. The authors then point out a known issue of structure determination techniques: structures of flexible loops are often missing, and these flexible parts may be important for the complex function. The authors suspect and then demonstrate with their studies that the extended loop region in Rrp42 protein is one such loop that is important for controlling the RNA pathway inside the exosome. They demonstrate that one loop in Rrp42 protein inhibits the entry of RNA directly into Rrp44 exonuclease site. Because of this loop, RNA enters this Rrp44 site only through the large barrel-like body of the exosome. The experiments are extensive and the results seem to be supported by the performed study. The identification of the above-mentioned extended loop as controlling the RNA pathway is a significant result. I recommend the paper for publication, after the authors consider the comments below.*

1) Figure S21 shows that RMSDs of open and closed Rrp42-EL (extended loop) both remain at < 4 Angstrom within 100 ns. The authors refer to this figure as showing the structures to be energetically stable. However, the RMSDs of the same two loops shown in Fig 5B are very different. Can the authors clarify the discrepancy?

We thank the reviewer to point out that our current phrasing and labeling is misleading. Figure 5B shows the RMSD of the C α atoms only of the extended loop of Rrp42 (Rrp42-EL) whereas the plot in Figure S21 reflects the deviation of all C α atoms of the complete simulation system (including the dynamic loop, that is very small compared to the complete system). While Fig S21 indicates that the whole exosome complex is globally equilibrated, Fig 5B indicates the local differences in dynamics related to Rrp42-EL. We clarified this in the captions of Fig. 5B and Fig. S21 as well as removed the misleading labels of Fig S21.

Fig 5B caption: “**Local** structural dynamics of Rrp42-EL in the closed (dark green) and open (cyan) state revealed by the RMSD of the C α carbons **only** of Rrp42-EL within the MD simulation.”

Fig S21 caption: “**Global C α root mean square deviation (RMSD)**. The plot of the RMSD of the C α carbon atoms **of the complete simulation system** over the course of the MD simulation of **the** Rrp42-EL closed (**A**) and open state (**B**) with respect to the MD starting structure converges towards a plateau.”

2) The results indicate that Exo9 has the Rrp42-EL 95% in the open state, whereas the same loop in Exo10 is 74% of time/population in the open state. The paper states that the loop closed state is presumed to be blocking the aberrant RNA pathway. Figure 6 confirms that kinetics of RNA degradation in Exo10 is different in Exo 10 complexes with and without the loop. How do the authors explain that the loop being only 26% of time in the closed state (or that only 26% of Exo10 complexes have closed loop when RNA is entering) is sufficient to block the aberrant RNA pathway? Is it reasonable to call the loop “flexible plug” (abstract) given the 26% closed state population?

Thank you for pointing this out. Indeed, an equilibrium with a major population of the loop in the closed state would seem more intuitive at first glance. However, we hypothesize that the population equilibrium and dynamics are fine-tuned to achieve a compromise between two conflicting objectives: Rrp42-EL in the closed state obstructs the path of properly inserted RNA to the active site. In order for this energetic barrier to not be rate-limiting, it should be as low as possible, i.e. the open state should be populated as much as possible (the residence time in the open state should be as long as possible). On the other hand, if the open state was to be fully populated, this would allow RNA to access the catalytic site through an aberrant path. To avoid this, the closed state should be populated as much as possible. An optimal balance would thus entail a population equilibrium and dynamics that obstruct the aberrant path, while keeping the energy barrier for properly insert RNA to pass the constriction site as low as possible. At present we cannot prove that with the observed population equilibrium (74% open) and dynamics ($k_{ex} \sim 35 \text{ s}^{-1}$) this optimal balance is achieved, however, we suspect that this is the case.

We amended the text and included the above line of reasoning on page 11, line 310-318: “We hypothesize that the observed population equilibrium and dynamics of Rrp42-EL (**Fig. 4, fig. S15**) are fine-tuned to achieve a compromise between two conflicting objectives: Rrp42-EL in the closed state obstructs the path of channel-bound RNA to the active site. In order for this energetic barrier to not be rate-limiting, it should be as low as possible, i.e. the *open* state should be populated as much as possible. On the other hand, if the open state was to be fully populated, this would allow RNA to access the catalytic site through an aberrant path. To avoid this, the *closed* state should be populated as much as possible. An optimal balance thus entails a population equilibrium and dynamics that obstruct the aberrant path, while keeping the energy barrier for channel-bound RNA to pass the constriction site as low as possible.”

3) At the end of page 7, it is stated that the closed state was obtained by interactively modeling Rrp42-EL into the unoccupied cavity near Rrp41. It would be good to explain in more detail what is meant by “unoccupied cavity” (density in cryo-EM or physical cavity?). The figures

show open space rather than a cavity where the loop is present. The SI description of modeling (pg S13) then mentions that Asp113/Rrp41 and Ala106/Rrp42 is experimentally expected to be 7 Angstrom, and the structure resulting from modeling has this distance at 14 Angstrom. What could be the error/deviation in the experimental result (7 Angstrom plus/minus what?), and what is the distribution of the specified distance in MD simulations (in the closed state)? Can the authors provide further discussion reflecting their confidence in the modeled structure truly representing the closed structure in experimental systems?

We thank the reviewer to show us that our wording of a cavity is misleading as it is actually a physically open space in the structural model of the Rrp42-EL open state. We rephrased this in the text accordingly. The difference between the experimental distance of 7Å and 14Å in the model is attributed to the fact that the experimental value is the distance of the spin label to D113tfmF and in our model between the C^α atoms of the residues (Asp113 Rrp42 and Ala106 Rrp42). Thus, only a qualitative comparison is possible. To clarify the comparison, we rephrased the related sentence in the main text and adopted Fig. S22 including its caption.

Page 10, lines 273-275: “In the closed conformation, the Rrp42-A106 C^α atom remains in close distance to the Rrp41-D113 C^α atom, whereas this distance is much longer in the open state (fig. S22, SI movie 1), which agrees qualitatively with the ¹⁹F PRE data (Fig. 4, fig. S18).”

SI page S65: “**Figure S22: Distances between the residues labeled for ¹⁹F PRE experiments.** (A) Location of the ¹⁹F and spin label within the modeled starting structures of Exo9 with open state (dark green) and closed state (cyan) Rrp42-EL. Rrp42 is shown in green, Rrp41 in red, Rrp45 in orange and Csl4 in blue. Zoom: Distance of the C^α carbon atoms of the labeled residues within the starting model of the MD simulation. (B) Measured distance between the C^α atoms of the residues labeled in the ¹⁹F PRE experiments (Rrp42-A106 and Rrp41-D113) within the MD simulation trajectories of the open (cyan) and closed state (dark green). Within the MD simulations neither the spin label nor the D113tfmF mutation are incorporated. The open and closed state show a clear difference in the C^α distances which is qualitatively comparable to the experimentally observed differences in the spin label to tfmF distance.”

The uncertainties of the distances obtained from the experimental data are stated in table S4 and we now changed the values mentioned on page S14, line 355 to 6.8±0.1 Å as determined from the experimental fit of a simple two state exchange model to the experimental NMR data.

Reviewer #2 (Remarks to the Author):

The article entitled “4D structural biology: quantitative dynamics in the eukaryotic RNA exosome complex” by Jobst et al. have combined crystallography, cryo-EM, Methyl-TROSY NMR, ¹⁹F-NMR, and MD simulation to understand the structural and dynamic mechanism of RNA exosome from the eukaryotic thermophile Chaetomium thermophilum (ctExo9). Current structures of exosome represent some snapshots of exosome function cycle. However, the extended loop region in Rrp42 (Rrp42-EL) and the entry loop in Rrp41 are largely unresolved in the current structures and it's unclear how the movement of these loop regions related to the exosome function. To address this fundamental and challenging question, the authors have used several different biophysical tools.

Particularly, ^{19}F -NMR revealed that Rrp42-EL samples closed and open conformations in both Exo9 and Exo10 complexes in the absence of RNA, while only the open conformation is sampled in the presence of RNA. Importantly, the authors also showed that the Rrp42-EL functions as a barrier to block an aberrant access path to the Rrp44 active site. Overall, the experiments are of high quality and the results are reasonably interpolated and the conclusion is clear. This is a great example how a collection of biophysical methods could complement each other to achieve deeper mechanistic understandings. However, I do have some questions and suggestions that may improve the manuscript.

1. In the “Main Text”, the introduction may need to be improved a little bit. I didn’t get what’s the authors initial motivation for the work and what particular questions the work is going to answer. There are Exo9 and Exo10 structures from yeast and human. Why the authors want to use crystallization and cryo-EM to get the structure of Exo9 from an eukaryotic thermophile? I think expanding the introduction would make readers better understand the work.

Thank you for this comment, it is important to be clear about the goals of a study. We rewrote the introduction section accordingly. Specifically we underline that the key goal of the study is to demonstrate that NMR can be employed on very large, asymmetric complexes, which were not amenable to NMR previously, and that a broad range of NMR methods can be applied to investigate such systems (page 3, line 47 – 61):

“Specifically, NMR enables the study of transient interactions and dynamics on a wide range of timescales, which are crucial for the function of enzymes, yet difficult to analyze by static structural methods. NMR can thus add the time dimension to the three structural dimensions that static methods provide.

Despite technological advances large and asymmetric complexes have eluded NMR investigation. Such complexes are substantially more challenging to study than large symmetric complexes, since symmetry gives rise to signal enhancement and spectral simplification. At the same time, large, asymmetric protein assemblies are ubiquitous (14) and thus of high interest to structural biology. Here, we demonstrate that by combining recent developments in labeling strategies and experiment design, very large, **fully asymmetric** protein complexes are amenable to NMR study. We show that a wide range of NMR experiments, informing on interactions, structure and dynamics can be strategically applied to such systems. In combination with cryo-EM, X-ray crystallography and molecular dynamics (MD) simulation the NMR experiments provide unique insights into *dynamic* properties of protein complexes.“

We also mention now in the introduction that we study the exosome of a eukaryotic thermophile because of its increased thermal stability as compared to exosomes from other organisms and the ability to express subunits individually and reconstitute the complex *in vitro*. Page 4, lines 79-83: “Here, we study the exosome of *Chaetomium thermophilum* since it exhibits improved thermal stability compared to exosome complexes from other organisms allowing for experiments to be conducted at up to 40 °C. In addition, *C. thermophilum* subunits can be expressed as monomers (or

heterodimers) with sufficiently high yields and the complex can be reconstituted *in vitro* rendering it suitable for NMR study.”

Even though it is expected that the overall structure of the exosome is conserved between yeast, humans and *C. thermophilum*, we deemed it necessary to verify this so as to not have to rely on structure prediction tools. Therefore, we solved the X-ray structure. Since the resolution of this structure was relatively low and with the goal of obtaining additional structural insights, we also decided to obtain the cryo-EM structure.

2. In the section “RNA displaces a channel exit loop”, the authors described methyl TROSY spectra first, but the later referred figures, Fig. 4B, C, F, G, are 19F-NMR, which should be the content of the next section “Rrp44 and RNA modulate the Rrp42-EL dynamics”. I guess the authors have made something wrong there. I guess the referred figures might be Fig. 3x? Please make it clear.

Thank you for pointing this out. The figure references were wrong. We replaced:

1. page 7, line 174: (Fig. 4B, C, F, G, fig. S12) by (Fig. 3B, C, F, G, fig. S12)
2. page 7, line 177: (Fig. 4D, fig. S12) by (Fig. 3D, fig. S12).

3. The complex has ribonucleolytic activity, how can the NMR spectra be recorded in the presence of RNA?

In the SI section on p. S7 line 189-191 we state that in the presence of RNA, the NMR experiments on Exo10 were conducted with inactive Rrp44 (Rrp44^{D168N-D536N}). For clarity, we now include this information also in the main text on page 6, line 149-151: “To prevent RNA degradation, we reconstituted Exo10 with inactive Rrp44^{D168N-D536N} in all NMR experiments conducted in the presence of RNA.”

4. In Fig S15, could you explain what the right panels of the CEST profiles are? In Fig S15A, the peaks of the two conformations coalesced together, for which the CEST should not work. Why the authors say it identified a minor state in Line 165?

We expanded on the description of the right panels of the CEST profiles in the caption of figure S15. These panels plot the asymmetry of the CEST dip, where 0 Hz is the position of the most intense resonance and thus the lowest value of the dip. On the x-axis the intensities at frequency offsets < 0 Hz are plotted against the intensities at corresponding frequency offsets > 0 Hz on the y-axis. If the data points are along a 45° line (e.g. S15 C), this implies that the CEST dip is perfectly symmetric and no exchange can be observed. If the the data points are not located on a 45° line (S15A, E) the CEST dip is asymmetric indicating that there is exchange. This allows to assess exchange dynamics even when there is no pronounced second (minor state) exchange dip but a situation that is close to coalescence of the minor and major state dips. This also means that CEST experiments can indeed be employed when dips coalesce, e.g. because of fast exchange, small $\Delta\omega$ or relatively strong B_1 field. This approach has previously been established by the Al-Hashimi

group, which they termed high-power CEST, and we now include a reference to their work (doi: 10.1002/anie.202000493, new reference 50), SI page 57:

“The symmetry of the main CEST dip is assessed by plotting intensities that are equidistant from the main dip (at 0 Hz) against each other, where offsets < 0 Hz are on the x-axis and the respective offsets > 0 Hz are on the y-axis. If all data points are located on a 45° line, the main dip is symmetric and the CEST experiment does not report on dynamics (as in **C**). If data points are not located on a 45° line, the main dip is asymmetric due to CEST-observable exchange dynamics (as in **A** and **E**). This approach allows to determine exchange dynamics even in the absence of a pronounced exchange dip (as in **A**) (50).”

Note, that the approach requires that 0 Hz offset was set to be at the maximum intensity of the resonance and that data points were acquired symmetrically around 0 Hz. We included a sentence in the method section stating that we ensured this was the case, p S8 line 216-217: “The central frequency (0 Hz) was set on resonance with the most intense fluorine resonance and frequency offsets were sampled symmetrically around 0 Hz.”

Indeed from the CEST profile alone we could not conclude that there is fast millisecond exchange or a minor population, because several parameters (B_1 , population, $\Delta\omega$ and k_{ex}) give rise to the observed profile. However, the data is consistent with fast millisecond exchange and a minor population. For Exo9, CPMG relaxation dispersion data clearly show the existence of a minor population in fast millisecond exchange. This conclusion is consistent with the fact that CEST profiles of the Exo9 resonance are asymmetric. Therefore, we rephrased line 186-188 on page 7.

“CPMG relaxation dispersion (**Fig. 4B, fig. S15B**) and chemical exchange saturation transfer (CEST) measurements (**fig. S15A**) reveal the presence of a second, minor conformation” was changed to “CPMG relaxation dispersion (**Fig. 4B, fig. S15B**) measurements reveal the presence of a second, minor conformation and, consistently, chemical exchange saturation (CEST) experiments show an asymmetric CEST dip (**fig. S15A**).”

5. I am a little concerned about the interpolation of the R1 PRE effect of Exo10-Rrp41D113tfmF in the absence of RNA (Figure 4F). The protein should sample multiple conformations (at least 2) as is proven in Exo9 and also in A106C-BTFA labeling. While the ^{19}F spectra showed a clear coalesced peak as Fig 4c for Exo9, the 1D- ^{19}F spectra of Exo10-Rrp41D113tfmF without RNA look like a partial overlapping of a broad peak with a sharp peak. The authors seem plotting the signal as single peak for characterizing the R1 PRE effect. Of note, the observed R1 PRE effect will be affected by the exchange rate and also the population of the two states (one features strong PRE, while another features weak PRE) as discussed in Nat Chem Biol 16, 1006–1012 (2020). <https://doi.org/10.1038/s41589-020-0561-6>. In the case of Exo9-No RNA, the conformation exchange rates are much faster than the R1 PRE, which should result in an averaged observed R1. That means, the real R1 PRE for the closed conformation will be dramatically larger, due to only 5% of closed conformation. In the case of Exo10-No RNA, the exchange rate is slowed down, but may still significantly large than the R1 PRE. I would suggest the authors try to deconvolute the spectra to see are the two peaks relax similarly. Lower the temperature may also help to shift the exchange rate and thus the R1 relaxation behaviors of the two peaks. In the manuscript, the authors didn't

consider such effect on the R1 PRE. Therefore, I would suggest to including this into the R1 PRE data analysis and discussion.

Apo Exo10 Rrp41^{D113tfmF} indeed displays a shoulder in the ¹⁹F spectrum, which indicates the existence of two conformations. As shown in fig. S9 A and B we did not observe any millisecond exchange dynamics. This suggests that the exchange between the two conformations, if it exists, must be slow.

As suggested by the reviewer we deconvoluted the T₁ data for diamagnetic Exo10 Rrp42^{C59S-A108C-TEMPO} Rrp41^{D113tfmF} in fig. S18C (no RNA). We did this by first deconvoluting the 1D spectra of Exo10 in the diamagnetic sample (see new fig. S18E). We find that the shoulder has a population of ~75%. We used the thus determined linewidths and peak positions to then deconvolute the T₁ data (see new fig. S18F, G). Within error, we obtain indistinguishable T₁ rates for both resonances. We include the analysis in figure S18 E-G. We do agree that a deconvolution of the resonance would be the correct way to perform the full analysis, however, the signal-to-noise ratio of the remainder of the data is insufficient to robustly do that. This means that the parameters that we obtain from the fits are averages of the two conformations of Rrp41^{D113tfmF}. We include this line of reasoning on page 9, line 242-247: “We further assumed that Rrp41^{D113tfmF} adopts only one conformation. For diamagnetic Exo10 Rrp42^{C59S-A108C-TEMPO} Rrp41^{D113tfmF} this is clearly not correct since the resonance displays a shoulder, implying that parameters extracted for this sample are population-weighted averages of the two conformations of Rrp41^{D113tfmF} in Exo10. We deconvoluted 1D spectra and T₁ data for this sample. While the shoulder peak (75% population) shows faster R₂ relaxation, both components have indistinguishable R₁ relaxation rates (**fig. S18 E-G**).”

We also include a description of the deconvolution approach in the method section, page S13, line 333-337: “**The 1D ¹⁹F spectrum and T₁ data of diamagnetic Exo10 Rrp42^{C59S-A108C-TEMPO} Rrp41^{D113tfmF} were deconvoluted** by fitting the 1D ¹⁹F spectrum to a double Lorentzian function. The thus obtained peak positions and linewidths were fixed for the deconvolution of T₁ data, for which only intensities of both peaks were parameters. Eq. S1 was fitted to those intensities to obtain R₁ relaxation rates for the deconvoluted resonances.”

We believe that our analysis remains valid since we do account for averaging of R₁ and R₂ relaxation rates that are caused by the exchange processes (fast ms motions in Exo9, slow ms motions in Exo10) as outlined in the method section. This is achieved by calculating Γ_1 and Γ_2 relaxation rates from spectral density functions where S^2 and τ_{loc} are parameters. Having thus calculated paramagnetic contributions to R₁ and R₂, we apply the Bloch-McConnell formalism to account for exchange dynamics and obtain intensities that we fit to our experimental data by varying the input parameters S^2 and τ_{loc} (see SI page S11-S13).

The reviewer is correct in their assessment of the relaxation rates of the open and closed states, which are presented in SI Table S4. For the closed state we observe dramatically increased R₁/R₂ relaxation rates while the PRE effect on R₁/R₂ relaxation rates of the open state is zero or almost zero. We also agree that the populations and dynamics will affect the averaging of the observed PRE rates and, as outlined in the previous paragraph, we do account for that in our model.

6. The authors may need to give references about how the order parameters are simulated.

The simulation of order parameters is described in the method section, page S11 – S13 and we reference the seminal work by Solomon/Bloembergen that connects order parameters to relaxation enhancements as well as the work by Clore and Iwahara that demonstrates how to combine R_2 relaxation enhancements with the model-free formalism and integrate that into the Bloch-McConnell formalism. We did not reference the model free formalism but include a reference now (page S12 line 308, reference 24: G. Lipari, A. Szabo, Model-free approach to the interpretation of nuclear magnetic resonance relaxation in macromolecules. 1. Theory and range of validity. *J. Am. Chem. Soc.* **104**, 4546–4559 (1982)). R_1 relaxation enhancements have previously been integrated in a similar way to account for slow exchange dynamics. The work was already referenced in the main text but we now also refer to it in the method section (page S11, line 298, reference 20: Y. Huang, X. Wang, G. Lv, A. M. Razavi, G. H. M. Huysmans, H. Weinstein, C. Bracken, D. Eliezer, O. Boudker, Use of paramagnetic ^{19}F NMR to monitor domain movement in a glutamate transporter homolog. *Nat. Chem. Biol.* **16**, 1006–1012 (2020)).

Reviewer #3 (Remarks to the Author):

Recommendation:

Accept with minor corrections

Liebau et al. present an excellent study highlighting the complementary nature of dynamic NMR experiments and MD simulation to static X-ray or cryo-EM structures, focusing on a eukaryotic RNA-degrading large complex as their representative target. Well-executed methyl and fluorine NMR experiments allow an in-depth investigation of various interacting subunit loops and their role in facilitating or preventing RNA exit and entry into the exosome. Their conclusions are clearly drawn from the presented evidence without far-reaching extrapolation with clear and intricate figures, with minor flaws in methodology and data presentation. The authors performed an immense amount of design, purification, and characterization work that is to be commended. The reviewers recommend the manuscript for publication with the following suggested minor changes for improvement.

Comments:

The reviewers recommend submitting all raw and processed NMR data to a database such as BMRBig (<https://bmrbig.org>) to facilitate open science practices.

Thank you for pointing this out. We agree that open science practices are important and submitted all raw and processed NMR data to BMRBig with the entry ID BMRbig108. We added this information in the data and materials availability section, p. 27, line 445-446: “Raw and processed NMR data have been submitted to BMRBig with the entry ID BMRbig108.”

The reviewers recommend changing the reference citation for the 2D methyl-TROSY via SOFAST-HMQC to the following, as the current reference, while utilizing a similar pulse

program, is not the correct one: o Amero et al. Fast two-dimensional NMR spectroscopy of high molecular weight protein assemblies. 2009. J. Am. Chem. Soc., 131(10), 3448-3449. <https://pubs.acs.org/doi/10.1021/ja809880p>

Thank you for the observation, we changed the reference accordingly (reference 8 in the main text, reference 14 in the SI).

The video file (Movie S1) associated with this manuscript submission could not be opened for evaluation. Please check compatibility with all OS.

Thank you for pointing out this problem. We converted the movie from mpg to mp4 format. In our hands, the mp4 movie runs on current installations of Windows, Mac and Linux (Ubuntu, Debian) provided appropriate software is installed.

The reviewers recommend bringing Supp Fig S26 Panel C to the main text and combining with Fig 6. In that same idea, we recommend removing Fig S26D (repetitive with Fig 6B), and potentially consider combining Fig S25, Fig S26A-C into Fig 6 if space allows.

We replaced Fig. 6 by fig. S26 and added new analyses to fig. S26, see next comment. For space reasons we decided to not move fig. S25 to the main text.

Please add how you determined the linear activity regime in the Supplemental Methods section near lines 428-431. The explanation is somewhat buried in the text in the caption for Figure S26. Visually, the excluded points look linear, and so further clarification of how you excluded those points is needed.

The reviewers are correct in pointing out a certain problem here. It is indeed not straightforward to identify the linear activity regime since the data is relatively sparse. Therefore, we also analyzed the data for alternative linear activity regimes by including/excluding further data points in the linear fit as outlined in a new table S10A and shown in Fig. 6 in the main text and fig. S26 in the SI. Catalytic rates are shown in table S10B. Note, that we changed the main text Fig 6 (analysis 1) to now include all data points for Exo10 Rrp45-L Rrp42^{Δ93-125}. Importantly, irrespective of the data points analyzed our conclusion, namely that the catalytic rates of wtExo10 and Exo10 Rrp42^{Δ93-125} are identical within error limits, while Exo10 Rrp45-L Rrp42^{Δ93-125} shows enhanced activity compared to Exo10 Rrp45-L, holds. We include this line of reasoning in the method section page S17, line 452 – 460:

“Since it is not straightforward to identify the linear activity regime, we also analyzed the data for alternative linear activity regimes by including/excluding further data points in the linear fit as outlined in table S10A and shown in Fig. 6 in the main text and fig. S26 below. Catalytic rates are shown in table S10B. Importantly, irrespective of the data points analyzed our conclusion, namely that the catalytic rates of wtExo10 and Exo10 Rrp42^{Δ93-125} are identical within error limits, while Exo10 Rrp45-L Rrp42^{Δ93-125} shows enhanced activity compared to Exo10 Rrp45-L, holds.”

The reviewers recommend clarifying the protein sequences used for CS1, CS2, and PIN constructs, as only the Rrp44 is denoted in the UniProt table (Supp Table S5).

Thank you for pointing out this omission. We included sequence information about the truncated Rrp44 constructs in Table S6D.

Please clarify what peaks on the SEC correspond to the pooled fractions for Exo9 & Exo10. There are 3 peaks evident in the chromatogram, one of which overlaps in both samples, but two that are shifted between Exo9 and Exo10. (Supp Fig S3B).

We modified the figure and its caption to clearly state which fractions were pooled and how we interpret the three elution peaks. SI page S42-S43, lines 538-540: “Fractions between the dashed green/black lines were pooled. Elution peak 1 corresponds to exosome aggregates, isolated exosome complexes give rise to elution peak 2 and excess subunits that did not form complexes give rise to elution peak 3.”

The first two peaks are shifted by virtue of the complex size (Exo10 vs Exo9 in aggregated and non-aggregated form). The third peak encompasses all excess subunits that can not be separated on a preparative S200 column.

The gel in Supp Fig S3C has additional, higher molecular weight, unlabeled bands between Rrp42 and Rrp44. Please clarify if you think these are oligomer bands of the subunits or possible degradation of the exosome, or a contaminant from the Rrp44 purification.

These bands are contaminants from Rrp44 purifications and we labeled the gel accordingly and modified the caption on page S43, lines 541-542: “* denotes impurities that arise from the purification of Rrp44.”

The main text mentions Rrp6 (line 59-60) but this protein is not included in the study, and does not clearly connect the preceding and proceeding sentences. The reviewers recommend removing the sentence to improve clarity and manage reader expectations.

We removed the reference to Rrp6.

The conclusion of the text feels almost separate from the results presented. The reviewers recommend, between the two sentences on lines 287-292, to quickly summarize the “quantitative and functionally important information” in the “invisible structures” gleaned from the exosome that are presented in the text.

We added a brief summary, page 12, lines 334 – 343:

“In particular, we showed that quantitative and functionally important information can be obtained for regions that are “invisible” in structures derived from cryo-EM and X-ray crystallography. We demonstrated this for an entrance loop of the exosome and for Rrp42-EL, both of which are not visible in static structural snapshots. The dynamics of Rrp42-EL are altered by Rrp44, which slows

down exchange dynamics by two orders of magnitude and increases the population of the closed state, and by RNA, in the presence of which only the open state is populated. In addition a combination of MD, methyl and ^{19}F NMR allowed us to characterize structural features of Rrp42-EL. The observed dynamics are of functional importance, since Rrp42-EL seals an aberrant access path to the catalytic site, while being sufficiently dynamic to allow passage of properly inserted RNA, a conclusion that could not have been reached from static structures alone.”

Parallel to the previous comment, the introduction lacks the central question or thesis statement you are trying to answer. It is contained within the abstract (“Our work thus demonstrates...thereby adding the time domain into structural biology.”). The reviewers recommend defining this 4th dimension aspect in the main text first paragraph, either in conjunction with or after the sentence “This thus opens ample opportunities...or in silico tools” to clearly connect the title, abstract, and project rationale.

We expanded the first paragraph (page 3, lines 47 – 50): “Specifically, NMR enables the study of transient interactions and dynamics on a wide range of timescales, which are crucial for the function of enzymes, yet difficult to analyze by static structural methods. NMR can thus add the time dimension to the three structural dimensions that static methods provide.”

In addition and also in response to reviewer #2 we now outline the research question in a new paragraph in the introduction section, page 3, lines 52 – 61: “Despite technological advances large and asymmetric complexes have eluded NMR investigation. Such complexes are substantially more challenging to study than large symmetric complexes, since symmetry gives rise to signal enhancement and spectral simplification. At the same time, large, asymmetric protein assemblies are ubiquitous (14) and thus of high interest to structural biology. Here, we demonstrate that by combining recent developments in labeling strategies and experiment design, very large, **fully asymmetric** protein complexes are amenable to NMR study. We show that a wide range of NMR experiments, informing on interactions, structure and dynamics can be strategically applied to such systems. In combination with cryo-EM, X-ray crystallography and molecular dynamics (MD) simulation the NMR experiments provide unique insights into *dynamic* properties of protein complexes.”

It is stated in the main text that “the spin-label gives rise to enhanced R1 and R2 relaxation rates...” (lines 119-121). However, Fig S8 shows only the PRE (T1) values, which are within standard deviation from each other (i.e. no statistically significant enhancement). Table S3A also shows only the R1/T1 values. Please include the R2 rates (if calculated) or adjust the text to reflect the qualitative data in Fig S8B.

We did not calculate R_2 relaxation rates for these experiments but draw the conclusion qualitatively based on Fig S8B. We altered the text to reflect the qualitative nature of the conclusion, page 6, lines 139 – 142: “The spin-label gives rise to enhanced R_1 relaxation rates (fig. S8, table S3A). R_2 relaxation rates are also enhanced as judged from a qualitative comparison of diamagnetic versus paramagnetic spectra (fig. S8). This establishes that the Rrp41 entry loop is located at the entry site of the RNA channel.”

We do not agree with the comment that R_1 rates are not enhanced. As shown in table S3A (referenced in line 140, page 6) the R_1 values of the paramagnetic and diamagnetic experiment are different within one SD, though admittedly barely ($1.3 \pm 0.1 \text{ s}^{-1}$ vs $1.0 \pm 0.1 \text{ s}^{-1}$). Γ_1 , which is calculated from those values, is indeed not significantly different between Exo9 in the presence and absence of RNA due to error propagation. To be more transparent about that, we also included R_1 relaxation rates in fig. S8 and added the above reasoning to the caption of fig. S8, page S8, lines 574-577: “ R_1 relaxation rates and Γ_1 PREs derived from Eq. S1 and Eq. S3 are indicated in the panels (see table S3A). The spin-label gives rise to enhanced R_1 rates in the absence of RNA. For Γ_1 the effect is not statistically significant due to error propagation.”

The sentence in lines 209-211 are confusing. Please rewrite or split the ideas to improve clarity, for example: “Our data thus imply that the invisible Rrp42-EL is more rigid in the closed conformation sampled more prominently by the Exo10 complex than in the ensemble of open conformations populated by the Exo9 complex.”

Thanks for pointing this out. We rewrote the sentences to (page 8, lines 233 – 235): “Our data thus imply that the invisible Rrp42-EL is more rigid in the closed conformation than in the ensemble of open conformations. At the same time the closed conformation is more prominently populated in the Exo10 complex than in the Exo9 complex”.

Please clarify if the RNA sequence is crucial for Exo10 function and the selection criteria of the 46-mer/80-mer comprised of just A/G bases used in the study (Supp Table S7).

To address this comment we included a sentence in the introduction section page 4, lines 74 – 75: “Isolated exosomes degrade single-stranded RNA irrespective of the nucleotide composition.”

We further included a sentence in the method section (page S6, lines 166 – 170) to justify the choice of the sequence: “To ensure that the RNA was single-stranded (i.e. devoid of hairpin or G-quadruplex structures), we designed RNAs consisting of random sequences of A and G nucleotides that are not expected to form any stable secondary structure elements. 80mer RNA was used for the activity assay to provide better read-outs, whereas a shorter 46mer RNA was employed in the NMR experiments to avoid possible multimerization of the exosome complex.”

It is stated in the main text that the Exo10 complex with Rrp42C59S, A106C-TFA retains ribonucleolytic activity, which is true, however the gel shows an enhanced activity compared to the WT (Fig S14) in contrast to the Fig S14 caption that states there is a “highly similar rate.” Please indicate why you think this rate appears increased compared to non-modified Rrp42.

We amended the caption and weakened the statement on page S55 (“similar rates” instead of “highly similar rates”) and added to the caption: “The difference in degradation rates observed is within the measurement uncertainty of this qualitative evaluation.”

Please clarify why the Rrp44 band so much stronger than the other subunits in Exo10 (Fig S3C) if the protein addition was stoichiometric (Supp Methods; lines 123-128) Additionally,

please address why the band intensity of shared subunits between Exo9 and Exo10 is vastly different (Fig S3C).

Regarding the first point (more intense band for Rrp44 than for the other components), we believe that the preparation is indeed stoichiometric. The SDS-PAGE gel shown in Figure S3 has been stained with Coomassie G-250, which binds primarily to basic and aromatic amino acids (Compton et al., 1985), and therefore stains different proteins with different intensity, roughly correlating with their size. Since Rrp44 is 2.5 times bigger than the next biggest component (Rrp42), its band appears much stronger than the others on the gel. Accordingly, the intensity of the bands corresponding to the other subunits also decreases according to their size (compare for example Rrp42 to Csl4). This is not as obvious in the Exo9 lane since the sample has been slightly overloaded. Please note that the presented gel looks similar to gels of exosomes published by other groups (compare for instance to <https://doi.org/10.7554/eLife.38686>, Fig 1A). In addition, the elution peak of free Rrp44 is separated by ~8ml from Exo10 and is thus visible in case of superstoichiometric addition. We include the mentioned reference in the caption of Fig. S3 on page S43, lines 543-544: “The gels suggest stoichiometric reconstitution of the complex and are analogous to previously published gels of *S. cerevisiae* exosome complexes (47).”

Regarding the second point, the band intensities between Exo9 and Exo10 differ because the protein concentrations are different in the gel sample. This information is indeed lost due to the normalization of the absorption traces in fig. S3B, the purpose of which is to show that Exo10 elutes earlier than Exo9 as expected. We now included this information in the caption, stating that the concentration of Exo9 is roughly twice the concentration of Exo10. Page S43, lines 542-543: “Note that the concentration of Exo9 is ca. 2x the concentration of Exo10 explaining stronger bands for the Exo9 gel.”

Figure edits:

Major

Fig 3 panels are mislabeled in the main text as Fig 4 panels (section “RNA displaces a channel exit loop” from lines 152-155).

We replaced:

1. page 7, line 174: (Fig. 4B, C, F, G, fig. S12) by (Fig. 3B, C, F, G, fig. S12)
2. page 7, line 177: (Fig. 4D, fig. S12) by (Fig. 3D, fig. S12).

Minor

Fig 1: The label “Rrp44/Dis3” is confusing, as Dis3 is the human name and is never utilized nor mentioned again. The reviewers recommend removing “Dis3” from the figure and just denoting the subunit complex as Rrp44.

We removed “Dis3” in the figure and its mentioning in the introduction section retaining only “Rrp44”.

Fig S7: Please enlarge the spectra.

We enlarged the spectra. In addition, we also enlarged spectra in fig. S4, S5, S6, S8 and S13 and the structures in fig. S17.

Fig S9: Please enlarge the graphs.

We enlarged the graphs.

Fig S10 (line 138 in main text): Please change the cyan to another color for easier reading (e.g. dark blue, purple).

We changed the color to dark blue.

Fig S11A (line 144): Fig S11 does not contain panel A, correct in text.

We corrected that.

Fig S12: The reviewers recommend putting small legend of orange = Ipara and black = Idia on the graph rather than in the caption, like in Fig S8A. Please enlarge the spectra.

We made the suggested changes.

Fig S16: Please change the cyan to another color (darker). Please enlarge the figure.

We changed the color to dark blue and enlarged the figure.

Reviewer #4 (Remarks to the Author):

In the work by Liebau et al., the authors used sophisticated NMR spectroscopy to analyze the structural dynamics of eukaryotic RNA exosome complex. Using the eukaryotic thermophile Chaetomium thermophilum exosome complex as a working model, the authors proved the methyl-group and fluorine NMR experiments can detect flexible or dynamic structural elements that are unavailable by crystallography or cryo-EM. Their NMR spectroscopy results agree with the previous determined structural studies of the eukaryotic RNA exosome complexes. But they identified an extra-loop of Rrp42 that appears to be a plug to block an aberrant route of RNA towards the active site of the exosome. This work is largely a demonstration of the state of the art of NMR spectroscopy on multi-subunit large complex by a tour-de-force work. The biological discovery is rather limited. Some specific questions for the authors to improve their work.

1. Previous works have shown that the Rrp44 interacts with Exo9 in at least two distinct conformations in the presence of various length of RNA substrates (for instance, 24 nt vs 12 nt). Have the authors studied the Rrp44's conformation change upon RNA binding and the interaction variation of Rrp41, Rrp45, Rrp42 with Rrp44 in the different RNA substrates?

Indeed, these are important questions that we are currently addressing. However, we believe that such experiments are beyond the scope of the present manuscript.

2. Have the authors measured the PRE effects of Rrp42-EL with IM-labeled Rrp44 and the effects in the presence of RNA substrates?

In this manuscript, we did not investigate Rrp44 itself. We believe that the suggested experiment is important since it could provide information about relative conformations of Rrp44 with respect to Rrp42-EL. However, as mentioned in response to comment #1 we believe that the dynamics of Rrp44 warrant a separate study and their investigation is beyond the scope of this manuscript.

3. The Rrp42 EL is not conserved among human, yeast and the thermophile exosome. Therefore, its potential role during RNA processing does not seem conserved. This reduces the biological relevance of the discovery on ctRrp42 EL. What could be the unique biological function of the ctRrp42 EL in a thermophile species?

Rrp42-EL is indeed not conserved in this very position and it might not be conserved in other thermophiles, so the specific role of the loop that we characterize here may be limited to the exosome of *C. thermophilum*. However, long disordered segments that are invisible for structural methods occur frequently in eukaryotes and there are certainly a number of additional disordered loops in the *C. thermophilum* exosome. For instance, as discussed in fig. S8 a long disordered loop covers the entrance site of the exosome channel and is displaced by RNA. While we have not investigated the function of this loop specifically, it is not far-fetched to assume that its dynamics fulfill some regulatory function as well. By characterizing Rrp42-EL we have described one mechanism – a flexible plug that prevents aberrant access – that long disordered segments can fulfill. This may be a recurring theme in other organisms. In that sense, we believe that the observation that disordered or invisible regions in protein complexes are functionally highly relevant is of broader and general biological relevance.

We amended the main text and included the above line of reasoning on page 11, lines 319 – 324: “As shown in fig. S11, Rrp42-EL is not conserved across species, suggesting that its specific role might be limited to the exosome of *C. thermophilum*. However, long disordered segments occur frequently in eukaryotes; the entrance loop of Rrp41 constitutes one additional example that we discussed in fig. S8. By characterizing Rrp42-EL we here describe one mechanism – a flexible plug that prevents aberrant access – that long disordered segments can fulfill. This may be a recurring theme in other organisms and thus of broader biological significance.”

4. The authors showed that Rrp45-L reduced the activity of the exosome, and channel-blocked Exo10 Rrp45-L with the Rrp42 Δ 93-125 recovered the exosome activity. This may imply the

central channel partly recovered without Rrp42-EL plug, which does not necessarily prove the barrier block model of the direct access route.

We believe that this explanation cannot be fully excluded but is very unlikely. If the reviewer's interpretation were correct, our general conclusion would be altered in the sense that Rrp42-EL is required for the structural integrity of the channel. However, if it was true that Rrp42^{Δ93-125} recovered the channel this would imply that a deletion of this flexible element would give rise to a substantial structural deformation of the entire complex widening the channel so much that RNA can pass through it. This raises the question of whether such a deformed complex would even be stable. However, we observe that the complex can be reconstituted and is gel-filtration stable.

Another indication that Rrp42^{Δ93-125} is unlikely to affect the integrity of the channel is that Rrp42^{Δ93-125} does not affect the activity compared to the wildtype.

Furthermore, Rrp42-EL exists mostly in the open conformation where it is remote from the rest of the complex as shown by our PRE data. It is hard to conceive how such a structural feature would stabilize the structural integrity of the RNA channel.

Finally, the same insertion in Rrp45 has been used to inactivate the *S. cerevisiae* exosome, where the Rrp42-EL plug is not conserved. If the two would need to cooperate in blocking exosome activity, as suggested by the reviewer, one would expect the channel-blocking insertion (Rrp45-L) to be not as effective in the *S. cerevisiae* exosome.

Reviewer #5 (Remarks to the Author):

Thank you for reviewing our work and providing feedback.

REVIEWERS' COMMENTS

Reviewer #1 (Remarks to the Author):

The authors addressed all the questions and issues and made the satisfactory corrections. I recommend the manuscript for publication.

Thank you.

Reviewer #2 (Remarks to the Author):

The revised manuscript addressed my concerns and I would recommend to publish on Nat. Comm. I have only one suggestion. It would be better to add in Methods or figure legends how the error bars are generated.

Thank you.

We now include details on how the error bars are generated in all relevant figure captions (Fig. 3 A, B, C, D; Fig. 4 B, C, F; Fig. 5 A, B, C; SI figures S8 A; S9; S20 B, D, F, H; S16 B; S18 A, C; S26).

Reviewer #3 (Remarks to the Author):

All the points raised by the reviewer have been correctly addressed. I recommend publication of the article without further modification.

Thank you.

Reviewer #4 (Remarks to the Author):

The authors responded to my comments adequately. As stated in my comments in the first round of review, this work is technically sounding but does not provide much biological insights, which the authors claimed to pursue in their future studies. I do not have further comments on this work.

Thank you.

Reviewer #5 (Remarks to the Author):

Thank you for that.